# ActiveVOO: Value of Observation Guided Active Knowledge Acquisition for Open-World Embodied Lifted Regression Planning

**Xiaotian Liu[1], Ali Pesaranghader[2], Jaehong Kim[3], Tanmana Sadhu[2],**
**Hyejeong Jeon[3], Scott Sanner[1,4]**

[1]University of Toronto, Toronto, Canada    [2]LG Electronics, Toronto AI Lab, Toronto, Canada
[3]LG Electronics, Seoul, South Korea    [4]Vector Institute for Artificial Intelligence, Toronto, Canada

xiaotian.liu@mail.utoronto.ca    ssanner@mie.utoronto.ca
{ali.pesaranghader, jaehong02.kim, tanmana.sadhu, hyejeong.jeon}@lge.com

## Abstract

The ability to actively acquire information is essential for open-world planning under partial observability and incomplete knowledge. However, most existing embodied AI systems either assume a known object category or rely on passive perception strategies that exhaustively gather object and relational information from the environment. Such a strategy becomes insufficient in visually complex open-world settings. For instance, a typical household may contain thousands of novel and uniquely configured objects, most of which are irrelevant to the agent's current task. Consequently, open-world agents must be capable of actively identifying and prioritizing task-relevant objects to enable efficient and goal-directed knowledge acquisition. In this work, we introduce ACTIVEVOO, a novel zero-shot framework for open-world embodied planning that emphasizes object-centric active knowledge acquisition. ACTIVEVOO employs lifted regression to generate compact, first-order subgoal descriptions that identify task-relevant objects, and provides a principled mechanism to quantify the utility of sensing actions based on commonsense priors derived from LLMs and VLMs. We evaluate ACTIVEVOO on the visual ALFWorld benchmark, where it achieves substantial improvements over existing LLM- and VLM-based planning approaches, notably outperforming VLMs fine-tuned on ALFWorld data. This work establishes a principled foundation for developing embodied agents capable of actively and efficiently acquiring knowledge to plan and act in open-world environments.

## 1 Introduction

Open-world embodied planning is challenging due to partial observability and incomplete task-relevant knowledge. In the open world, knowledge acquisition becomes essential, as agents must identify and obtain relevant information necessary to generate feasible plans. Existing embodied AI systems often assume known object categories and rely on passive strategies that exhaustively collect objects and relational information without contextual relevance [Camoriano et al., 2017]. These approaches quickly become infeasible in complex, information-rich environments due to a combinatorial explosion of objects and their relations to each other.

*Active knowledge acquisition* offers a better alternative by enabling the agent to proactively seek information that is relevant to the task at hand. However, existing work in this area has largely focused on locating or recognizing predefined object classes [Zhu et al., 2017, Yang et al., 2018] or information gain [Zurbrügg et al., 2022], thereby overlooking a crucial preceding reasoning step: identifying which objects are actually relevant to the goal. To illustrate these challenges, Figure 1 shows a comparative example of active vs. passive knowledge acquisition.

39th Conference on Neural Information Processing Systems (NeurIPS 2025).

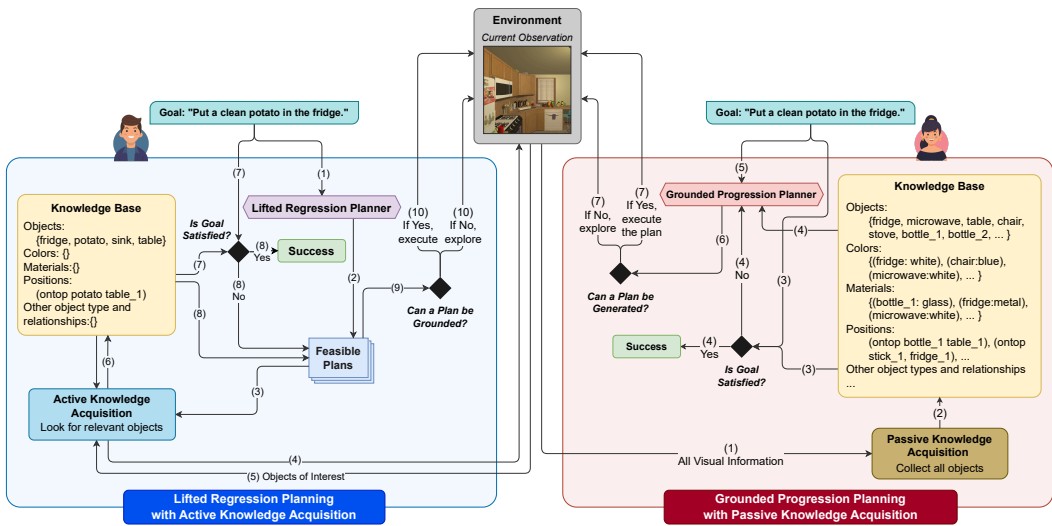

Figure 1: **Active vs. Passive Knowledge Acquisition.** The left panel illustrates the **active approach**, where lifted regression generates abstract subgoals that guide targeted VLM queries for object discovery. The numbered arrows indicate the sequential order of reasoning and sensing steps. In the active flow: 1) lifted subgoals are regressed from the goal, 2) candidate objects and relations are extracted for subgoals, 3) only relevant objects are sensed and grounded to update the agent's knowledge base, and 4) the agent explores the environment based on its sensing target. This closed loop of reasoning and sensing continues until a plan can be executed. The right panel depicts the **passive approach**, where a grounded progression planner relies on exhaustive object collection. Here, 1) the agent first gathers all visible objects, 2) attempts plan grounding, 3) if grounding fails due to missing objects, it re-enters exploration to collect additional data, and 4) Recovery occurs through further exploration (potentially guided by the failed plan context), but this process remains inefficient in open-world settings containing unbounded numbers of objects and relations.

In this work, we introduce ACTIVEVOO, an open-world planning framework with active knowledge acquisition. ACTIVEVOO adopts an object-centric view of knowledge, and decomposes active knowledge acquisition into two sub-tasks: 1) The agent determines which types of objects and relational properties are relevant to the goal. For example, it should recognize that a kettle is relevant when preparing a hot drink, while a TV remote is not. 2) The agent quantifies and prioritizes among the relevant objects to sense. For instance, when preparing breakfast, the agent must decide whether locating eggs is more important than finding a baking tray.

Object relevance can be determined by evaluating whether an object is necessary for executing a feasible plan. However, generating plans in open-world settings is feasible with typical grounded progression [Helmert, 2006] methods, which typically assumes the Closed World Assumption (CWA) and a fully specified initial state. An alternative is regression-based planning, where the agent searches backward from the goal to identify preceding subgoals without requiring explicit knowledge of the initial state [Reiter, 2001, Sanner and Boutilier, 2009]. Moreover, regression can be conducted at the lifted level, allowing compact relational reasoning that abstracts away from specific object instances.

ACTIVEVOO leverages lifted regression [Sanner and Boutilier, 2009, Ghallab et al., 2004] to obtain a set of relevant object descriptions, which it then quantifies and prioritizes based on how "helpful" each is to achieving the overall objective. For example, when tasked with "prepare a steak dinner," the agent must weigh the utility of searching for a steak versus locating a wine glass. To this end, we propose *Value of Observation (VOO)* to formalize active knowledge acquisition as a decision-making process that selects utility-maximizing objects to sense. VOO measures the expected improvement in utility obtained by observing a specific object, relative to the utility of acting without attempting to sense the object.

We evaluate ACTIVEVOO on the ALFWorld benchmark [Shridhar et al., 2020], which provides a diverse set of tasks in both visual and textual modalities. While prior work has primarily focused on the textual setting, the visual version of ALFWorld remains highly challenging due to partial observability, object diversity, and complex interaction dynamics. We demonstrate that ACTIVEVOO achieves significant improvements against the visual ALFWorld tasks, notably outperforming vision-

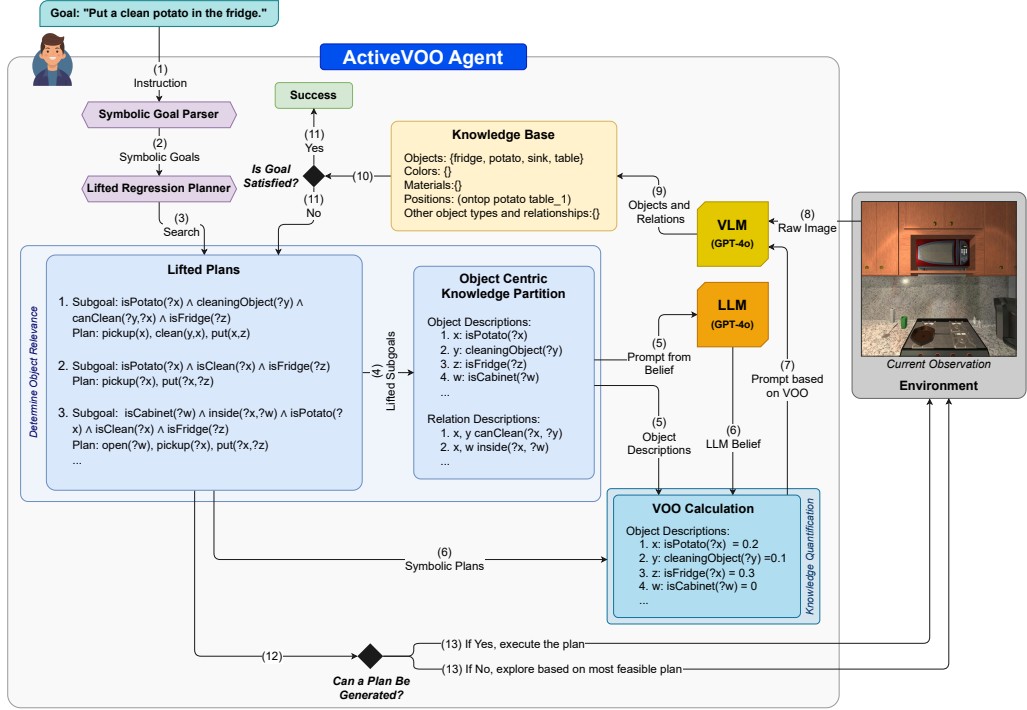

Figure 2: **Overview of the ACTIVEVOO Framework.** Figure 2 illustrates the full workflow of our approach, which integrates lifted regression, object–relation extraction, and Value of Observation (VOO)-guided active sensing. 1) Lifted regression decomposes the goal into a set of lifted subgoals that describe feasible abstract world configurations. 2) From each subgoal, object and relational descriptions are extracted. 3) The agent computes the VOO for each candidate object or relation based on the current KB. 4) Objects with high VOO are selected to guide exploration and sensing. This active cycle continues until all predicates in at least one lifted subgoal are satisfied, at which point the associated plan is instantiated and executed.

language models that are fine-tuned specifically for this benchmark. Figure 2 shows the overall workflow of ACTIVEVOO. To summarize, we make the following key contributions in this work:

- We enable active knowledge acquisition by extracting relevant high-level object and relational descriptions through lifted regression for open-world planning.
- We propose a decision-theoretic Value of Observation (VOO) approach that quantifies the utility of knowledge acquisition using commonsense beliefs from LLMs and VLMs.
- We achieve state-of-the-art performance on the visual ALFWorld benchmark in a zero-shot manner, outperforming even fine-tuned VLM baselines.

## 2 Methodology

A key challenge for open-world embodied agents is determining which *objects* are relevant to the task prior to planning, as novel and previously unknown objects may appear in the environment. Furthermore, exhaustively enumerating all relational information the agent observes is inefficient, since most objects the agent encounters are irrelevant to the task. To address this challenge, we propose a framework that identifies and prioritizes task-relevant objects through the following steps: 1) The agent applies *lifted regression* to derive subgoals representing possible world configurations where feasible plans can be executed; 2) it then extracts high-level (unary predicate) object descriptions from each subgoal; 3) for each identified object description, the agent estimates the probability of successful sensing by leveraging LLM-based commonsense; 4) the agent computes the *Value of Observation* (VOO) to prioritize object sensing based on expected utility gain; and 5) guided by VOO,

the agent actively senses and collects object information, updating its knowledge base and executing the plan once all relations in a subgoal can be satisfied.

An overview of ACTIVEVOO is shown in Figure 2. We assume the agent is equipped with high-level actions defined in PDDL [Haslum et al., 2019] syntax, an extension of STRIPS [Fikes and Nilsson, 1971], but lacks complete knowledge of object types and their relational states. The full action model for our experimental domain is provided in Appendix A; the examples below are modified for purposes of compact exposition. The agent receives a high-level natural-language instruction without step-by-step guidance and employs goal-conversion methods from [Song et al., 2023, Xie et al., 2023] to translate natural-language goals into PDDL form through few-shot prompting.

## 2.1 Lifted Plan and Subgoal Derivation

One approach for generating plans in the *open world* (i.e., an open domain of potential objects) is *lifted regression* [Sanner and Boutilier, 2009], which allows the agent to derive high-level plans and their necessary execution conditions without relying on concrete object instances. This is achieved by reasoning in a lifted representation and regressing backward from the goal to infer the abstract preconditions required for plan feasibility. By planning with lifted representations parameterized by variables, the agent reasons in high-level relational descriptions of objects rather than specific instances. For example, instead of concrete object instances such as "`oven`" or "`sink`", lifting enables the agent to reason about object properties and relations like `isClean(?x)` and `canClean(?y, ?x)`, deferring the instantiation of these variables until the relevant objects are observed. Lifted regression plans backward by successively regressing the goal through the action model to determine the abstract preconditions (i.e., subgoals) that must hold for executing a particular action to achieve that goal.

---

**Example (1)**   Suppose the agent's goal is to *put a clean potato in the fridge*, represented symbolically as

$$G := \texttt{isPotato(?x)} \land \texttt{isClean(?x)} \land \texttt{isFridge(?z)} \land \texttt{inside(?x, ?z)}.$$

Consider the action model

$$\texttt{PutIn(?x, ?z)} := \begin{cases} \textbf{Preconditions: } \texttt{holds(?x)} \\ \textbf{Effects: } \texttt{inside(?x, ?z)} \end{cases}$$

Regressing the goal $G$ through `PutIn(?x, ?z)` replaces the achieved effect `inside(?x ?z)` with the action's preconditions, yielding the regressed subgoal

$$g_1 := \texttt{isPotato(?x)} \land \texttt{isClean(?x)} \land \texttt{isFridge(?z)} \land \texttt{holds(?x)}.$$

This expression states that the agent has to be holding a potato `?x` that has already been cleaned and has found a fridge `?z` to complete the overall goal.

---

**Lifted Backward Search.**   Lifted regression enables the agent to reason backward from the goal to produce a set of high-level conditional plans capable of achieving the agent's objective under different possible environment configurations. In this work, we apply regression via *Lifted Backward Search* [Ghallab et al., 2004], adapted to open-world settings [Liu et al., 2025], to generate a set of conditional plans $\Pi = \{(g_1, \pi_1), \ldots, (g_n, \pi_n)\}$, where each lifted subgoal $g_i$ specifies a set of *minimal preconditions* that must hold for the corresponding plan $\pi_i$ to be executable. The details of the open-world lifted regression algorithm are provided in Appendix B.

---

**Example (2)**   Following the same setup as the previous example, suppose lifted regression yields two abstract subgoals that represent feasible world configurations.

$$g_1 := \texttt{isPotato(?x)} \land \texttt{isClean(?x)} \land \texttt{isFridge(?z)} \land \texttt{holds(?x)},$$
$$g_2 := \texttt{isPotato(?w)} \land \texttt{cleaningObject(?y)} \land \texttt{canClean(?y, ?w)} \land \texttt{isFridge(?z)}.$$

Each subgoal $g_i$ is associated with a lifted conditional plan $\pi_i$, which can be executed when the corresponding $g_i$ is satisfied:

$$\pi_1 : \big[\texttt{PutIn(?x, ?z)}\big],$$
$$\pi_2 : \big[\texttt{Pickup(?w), Clean(?y, ?w), PutIn(?w, ?z)}\big]$$

---

## 2.2 Object and Relation Description Extraction

**Extracting Object Descriptions.** To identify which objects are relevant for executing a plan, we first extract their symbolic descriptions from the lifted subgoals. We assume that an object's type and characteristics are captured by the unary predicates present in a subgoal formula $g$. For every variable $v \in \text{Vars}(g)$, we construct a conjunctive formula $o_{v,g}(v)$ that represents the properties associated with that object, and collectively define the set of all such object formulas as $\mathcal{O}$,

$$o_{v,g}(v) \equiv \bigwedge_{p(v) \in g,\ \text{arity}(p)=1} p(v), \qquad \mathcal{O} = \{\, o_{v,g}(v) \mid g \in \mathcal{G},\ v \in \text{Vars}(g) \,\}, \qquad (1)$$

where duplicates are removed via $\alpha$-renaming of variable symbols. This compact formulation encodes all unary predicates applied to each variable within $g$, yielding the object-level descriptions used as sensing targets during active knowledge acquisition.

---

**Example (3)** Continuing from the previous example, we extract unary predicates for each variable to form their corresponding object descriptions using Eq. 1. For $g_1$, we obtain:

$$o_{x,g_1}(?x) \equiv \texttt{isPotato(?x)} \wedge \texttt{isClean(?x)} \wedge \texttt{holds(?x)}, \quad o_{z,g_1}(?z) \equiv \texttt{isFridge(?z)}.$$

Similarly, for $g_2$, the extracted object descriptions are:

$$o_{w,g_2}(?w) \equiv \texttt{isPotato(?w)}, \quad o_{y,g_2}(?y) \equiv \texttt{cleaningObj(?y)}, \quad o_{z,g_2}(?z) \equiv \texttt{isFridge(?z)}.$$

After removing duplicate objects, we obtain the set of four object descriptions used in subgoals $g_1$ and $g_2$:

$$\mathcal{O} = \begin{aligned} &\{\texttt{isPotato(?w)}, \texttt{isPotato(?x)} \wedge \texttt{isClean(?x)} \wedge \texttt{holds(?x)}, \\ &\ \ \texttt{cleaningObj(?y)}, \texttt{isFridge(?z)}\}. \end{aligned}$$

---

**Extracting Relational Descriptions.** In addition to unary object properties, each subgoal formula $g$ may contain relational predicates that describe dependencies among multiple objects—for instance, for variables $\mathbf{v} = (\texttt{?y}, \texttt{?w})$, one may ask whether the relation $\texttt{canClean(?y, ?w)}$ holds. More precisely, for every relational predicate $p(\mathbf{v})$ with $\mathbf{v} = (v_1, \ldots, v_k)$ and $k \geq 2$ appearing in $g$, we construct a *relational formula* that links the corresponding object formulas, and collectively define the set of all such relational formulas as $\mathcal{R}$,

$$\rho_{\mathbf{v},g}(\mathbf{v}) \equiv \left( \bigwedge_{i=1}^{k} o_{v_i,g}(v_i) \wedge p(\mathbf{v}) \right), \qquad \mathcal{R} = \{ \rho_{\mathbf{v},g}(\mathbf{v}) \mid g \in \mathcal{G},\ p(\mathbf{v}) \in g,\ |\mathbf{v}| \geq 2 \}. \quad (2)$$

where each $\rho$ is an open first-order formula whose free variables correspond to the participating objects; it evaluates to true under an assignment $\mathbf{v}$ iff each $v_i$ instantiates an object satisfying its unary description and the relational predicate $p$ holds among them. Together, $\mathcal{O}$ and $\mathcal{R}$ form a lifted representation that links individual object descriptions to their relational structure for active knowledge acquisition.

---

**Example (4)** Continuing from the previous example, the subgoal $g_2$ contains one relational predicate, $\texttt{canClean(?y, ?w)}$. Using Eq. 2, we construct the corresponding relational formula:

$$\rho_{(y,w),\,g_2}(?y, ?w) \equiv \big( \texttt{cleaningObj(?y)} \wedge \texttt{isPotato(?w)} \wedge \texttt{canClean(?y, ?w)} \big).$$

This formula specifies that the relation $\texttt{canClean(?y, ?w)}$ needs to hold for object $\texttt{isPotato(?w)}$ and $\texttt{cleaningObject(?y)}$. We can then define the (singleton) set of relation descriptions for subgoal $g_2$:

$$\mathcal{R} = \{\texttt{cleaningObject(?y)} \wedge \texttt{isPotato(?w)} \wedge \texttt{canClean(?y, ?w)}\},$$

---

## 2.3 Sensing Belief Estimation via LLM Commonsense Probability Elicitation

Building on prior work showing that LLM token logits can approximate commonsense belief probabilities [Liu et al., 2023a, Chen et al., 2023], we estimate how likely it is for the agent to sense particular objects. For an object description $q_1(x) \wedge \cdots \wedge q_m(x)$, we query the LLM with statement $\mathbf{s}$: "Does there exist an object that is $q_1, \ldots, q_m$?" For a relational description $\rho$ over $\mathbf{v} = (v_1, \ldots, v_k)$ with predicate $p$, we query $\mathbf{s}$: "Can $p(\mathbf{v})$ hold given v such that $q_1(v_1), \ldots, q_k(v_k)$?" Then, given a

fixed context $\mathbf{c}$ (e.g., "You are reasoning about a typical household kitchen.") and a statement $\mathbf{s}$, we can obtain model-specific logits of "Yes" and "No" for $\mathbf{s}$ given that context $\mathbf{c}$ holds, denoted as

$$z_{\mathsf{Yes}}(\mathbf{c}, \mathbf{s}) := \mathrm{logit}_{\mathrm{LLM}}[\mathbf{c}, \mathbf{s} \rightarrow \text{"Yes"}], \qquad z_{\mathsf{No}}(\mathbf{c}, \mathbf{s}) := \mathrm{logit}_{\mathrm{LLM}}[\mathbf{c}, \mathbf{s} \rightarrow \text{"No"}], \qquad (3)$$

and apply the definition of conditional probability to estimate whether $\mathbf{s}$ is true when context $\mathbf{c}$ is true:

$$P(\mathbf{s} = T | \mathbf{c} = T) = \frac{P(\mathbf{s} = T, \mathbf{c} = T)}{P(\mathbf{s} = F, \mathbf{c} = T) + P(s = T, \mathbf{c} = T)} \approx \frac{\exp(z_{\mathsf{Yes}}(\mathbf{c}, \mathbf{s}))}{\exp(z_{\mathsf{Yes}}(\mathbf{c}, \mathbf{s})) + \exp(z_{\mathsf{No}}(\mathbf{c}, \mathbf{s}))},$$
$$(4)$$

Since some logit APIs are stochastic, we average $k$ logits across samples ($k$=5 in our case). For each $o \in \mathcal{O}$ and $\rho \in \mathcal{R}$, we obtain $p_o := \mathrm{Pr}(\mathbf{s}_o = T | \mathbf{c} = T)$ and $p_\rho := \mathrm{Pr}(\mathbf{s}_\rho = T | \mathbf{c} = T)$ which we use to independently estimate the probability of whether objects satisfying $\mathbf{s}_o$ or relationships satisfying $\mathbf{s}_\rho$ can be sensed in context $\mathbf{c}$. Prompt templates and paraphrasing details are provided in Appendix E.

## 2.4 Value of Observation (VOO) Computation

After extracting object and relational descriptions from lifted subgoals, the agent must determine *how to prioritize among candidate objects for sensing*. To this end, we propose *Value of Observation (VOO)*, a utility-based measure that quantifies the expected improvement in the agent's ability to achieve its goal with respect to sensing object candidates fitting specific descriptions. By evaluating the VOO for each potential sensing target, the agent can prioritize observations that are most informative, thereby enabling goal-directed and object-centric knowledge acquisition in open-world environments.

**Problem Setup.** Given the sets of object descriptions $\mathcal{O}$ and relational descriptions $\mathcal{R}$, we define the agent's knowledge states as a set of sensing beliefs over objects and relations:

$$S = \Big\{ \underbrace{(O^+, O^-, R^+, R^-)}_{s} \mid O^\pm \subseteq \mathcal{O}, \ R^\pm \subseteq \mathcal{R}, \ O^+ \cap O^- = \varnothing, \ R^+ \cap R^- = \varnothing \Big\}. \quad (5)$$

Here, $O^+$ and $R^+$ denote the sets of object and relational descriptions that the agent believes can be sensed, whereas $O^-$ and $R^-$ represent those that the agent believes cannot be sensed. Objects that are not included in either of these sets but are present in the overall object set $\mathcal{O}$ are considered *unknown* to the agent. The agent can perform a sensing action $a_o$ for an object description $o \in \mathcal{O}$, which transitions it into one of two possible successor states depending on whether $o$ can be sensed:

$$s'_+ = (O^+ \cup \{o\}, O^-, R^+, R^-), \qquad s'_- = (O^+, O^- \cup \{o\}, R^+, R^-), \qquad (6)$$

The two outcomes occur with respective probabilities $p_o$ and $1 - p_o$ estimated in Section 2.3. We formalize this knowledge acquisition process as a belief-state Markov Decision Process (MDP), whose full specification, including transition dynamics and reward formulation, is provided in Appendix D.

**Utility Estimation and VOO Calculation.** We define a utility function $U(s)$ reflecting the agent's belief that *any* subgoal $g \in \mathcal{G}$ is satisfiable given the agent's knowledge state $s$. For each subgoal $g$, we first identify the sets of *still-unknown* object and relational descriptions associated with $g$ as

$$\mathrm{obj}_{\mathrm{unk}}(g, s) = \mathrm{obj}(g) \setminus (O^+ \cup O^-), \qquad \mathrm{rel}_{\mathrm{unk}}(g, s) = \mathrm{rel}(g) \setminus (R^+ \cup R^-). \qquad (7)$$

Here, $\mathrm{obj}(g)$ and $\mathrm{rel}(g)$ denote the sets of object and relational descriptions associated with subgoal $g$, respectively.

---

**Example (5)** Continuing from the previous example, suppose the agent's current knowledge state is $s = (O^+, O^-, R^+, R^-)$, where $O^+ = \{\texttt{isPotato(?w)}\}$ and $R^+, O^-, R^- = \varnothing$, indicating that only the potato has been identified. Recalling previous subgoal

$g_2 := \texttt{isPotato(?w)} \land \texttt{cleaningObject(?y)} \land \texttt{canClean(?y, ?w)} \land \texttt{isFridge(?z)},$

the unknown object and relational descriptions are respectively:

$\mathrm{obj}_{\mathrm{unk}}(g_2, s) = \{\texttt{cleaningObject(?y)}, \texttt{isFridge(?z)}\},$

$\mathrm{rel}_{\mathrm{unk}}(g_2, s) = \{\texttt{cleaningObject(?y)} \land \texttt{isPotato(?w)} \land \texttt{canClean(?y, ?w)}\}.$

---

Assuming independent sensing outcomes with success probabilities $\{p_o\}_{o\in\mathcal{O}}$ for object descriptions and $\{p_\rho\}_{\rho\in\mathcal{R}}$ for relational descriptions, we define the belief probability a subgoal $g$ can be satisfied after sensing all currently unknown descriptions as:

$$\Pr\big(g \text{ is satisfiable} \mid s\big) \;=\; \prod_{o\in\mathrm{obj}_{\mathrm{unk}}(g,s)} p_o \cdot \prod_{\rho\in\mathrm{rel}_{\mathrm{unk}}(g,s)} p_\rho. \tag{8}$$

The utility of a given knowledge state $s$ is then defined as the maximum probability of any subgoal $g$ that can be made satisfiable from $s$.

$$U(s) \;=\; \max_{g\in\mathcal{G}} \left\{ \prod_{o\in\mathrm{obj}_{\mathrm{unk}}(g,s)} p_o \cdot \prod_{\rho\in\mathrm{rel}_{\mathrm{unk}}(g,s)} p_\rho \right\}. \tag{9}$$

---

**Example (6)** Continuing from the previous example, assuming independent sensing outcomes with success probabilities $p_{\texttt{cleaningObject}} = 0.9$, $p_{\texttt{isFridge}} = 0.8$, and $p_{\texttt{canClean}} = 0.7$, the belief that $g_2$ can be satisfied is
$$\Pr(g_2 \text{ is satisfiable} \mid s) = 0.9 \times 0.8 \times 0.7 = 0.504$$
For $g_1 = \texttt{isPotato(?x)} \wedge \texttt{isClean(?x)} \wedge \texttt{isFridge(?z)} \wedge \texttt{holds(?x)}$, the unknown descriptions are $\mathrm{obj}_{\mathrm{unk}}(g_1, s) = \{\texttt{isFridge(?z)}, \texttt{isPotato(?x)} \wedge \texttt{isClean(?x)} \wedge \texttt{holds(?x)}\}$, with probabilities $p_{\texttt{isFridge}} = 0.8$ and $p_{\texttt{isPotato} \wedge \texttt{isClean} \wedge \texttt{holds}} = 0.1$ giving

$$\Pr(g_1 \text{ is satisfiable} \mid s) = 0.8 \times 0.1 = 0.08$$

Using Eq. 9, the utility of the current knowledge state $s$ is defined as

$$U(s) \;=\; \max_{g\in\mathcal{G}} \left\{ \prod_{o\in\mathrm{obj}_{\mathrm{unk}}(g,s)} p_o \cdot \prod_{\rho\in\mathrm{rel}_{\mathrm{unk}}(g,s)} p_\rho \right\} \;=\; \max(0.504, 0.08) = 0.504$$

---

**Value of Observation.** Given utility $U(s)$, when the agent attempts to sense object description $o \in \mathcal{O}$, the expected utility of the outcomes of sensing vs. not sensing $o$ can be calculated as follows:

$$U_o^{\mathrm{sense}}(s) = p_o\,U(s'_+) + (1 - p_o)\,U(s'_-), \qquad U_o^{\mathrm{not\_sense}}(s) = U(s'_-). \tag{10}$$

Here, $s'_+$ and $s'_-$ denote the successor knowledge states corresponding to successful and unsuccessful sensing outcomes, respectively. The left equation represents the expected utility from sensing $o$, while the right shows the baseline utility when the agent does not sense $o$ and cannot make use of it. Then, the *Value of Observation (VOO)* quantifies the expected utility gain from actively sensing $o$:

$$\mathrm{VOO}_o(s) = U_o^{\mathrm{sense}}(s) - U_o^{\mathrm{not\_sense}}(s) = p_o\big[U(s'_+) - U(s'_-)\big]. \tag{11}$$

VOO thus measures how much the agent's expected progress toward a satisfiable subgoal improves by actively acquiring knowledge about a specific object.

While the astute reader may observe a non-incidental similarity between VOO and Value of Information (VOI) [Howard, 1966], there are causal differences in the computations. In VOI, deciding whether or not to make an observation (weather report) does not causally affect the outcome (whether it will rain). In contrast for VOO, an object must be actively observed to causally affect the outcome.

## 2.5 Active Knowledge Acquisition and Plan Execution

The ACTIVEVOO agent acquires knowledge from both the physical environment, by navigating to new locations and collecting visual observations, and by issuing language or vision–language queries that infer object relations using LLMs or VLMs. At each step, the agent selects the object description with the highest Value of Observation, $o^\star = \arg\max_{o\in\mathcal{O}} \mathrm{VOO}_o(s)$, and treats it as the next *exploration target*. It then queries an LLM to obtain the optimal location the agent can explore based on the exploration target. After navigating to the corresponding region, the agent captures an RGB frame and queries the VLM to detect instances that satisfy the top-$k$ (in our case, top 5) object descriptions ranked via VOO values. Successful detections are updated into the agent's knowledge state $s$, which in turn updates the estimated utilities $U(s)$ and all VOO scores. The agent also infers relational predicates (via LLM reasoning or direct observation) whose arguments correspond to the sensed object types. Once every predicate in a lifted subgoal $g \in \mathcal{G}$ is grounded, the agent instantiates

the associated plan $\pi_g$ and executes the corresponding plan. ACTIVEVOO enables the agent to allocate its limited sensing budget toward information that maximizes expected utility, seamlessly integrating physical exploration with commonsense reasoning over object relations. The complete execution flow of the agent is illustrated in Figure 2, and detailed implementation steps are provided in Appendix F of the supplementary material.

# 3 Experiments

**Experimental Setup.** We evaluate ACTIVEVOO on the ALFWorld benchmark using raw visual inputs in a zero-shot setting, without additional data for training or fine-tuning. We compare ACTIVEVOO against state-of-the-art LLM/VLM-based planners using both vision and language observations, including zero-shot methods as well as fine-tuned models on ALFWorld data. The performance is evaluated using Success Rate, defined as the agent's ability to complete the task in 30 steps. We average success rate over 3 runs. We set the regression length to 10 and the LLM temperature to 0.1, except when computing belief probabilities, where the temperature is set to 0. All experiments are carried out on a `6-Core AMD CPU` with `32GB of RAM`, and we rely on API calls for LLMs/VLMs. For ACTIVEVOO, we use `GPT-4o` both for belief probability estimation and as a vision-language model to extract relevant objects. We choose ALFWorld [Shridhar et al., 2020] as our benchmark to evaluate both text and vision observations modalities.

**Baselines.** We compare ACTIVEVOO against a diverse set of baselines across three categories: **vision only models**, **language only models**, and **vision language models**. We include ResNet-18 [Shridhar et al., 2020] and MCNN-FPN [Shridhar et al., 2020], which are standard computer vision architectures that take an image as input and output an action. For language-based models, we focus on prompting-based methods, REACT [Yao et al., 2022], REFLEXION [Shinn et al., 2024], DEPS [Wang et al., 2023], and AUTOGEN [Wu et al., 2023], which rely solely on LLMs for reasoning and planning with few-shot examples. Vision language agents take both raw pixel observations and natural language instructions as input, requiring multimodal reasoning. We include models that require data collection and fine-tuning, which are BUTLER [Shridhar et al., 2020], MINIGPT-4 [Zhu et al., 2023a], BLIP-2 [Li et al., 2023], LLAMA-ADAPTOR [Gao et al., 2023], INSTRUCTBLIP [Panagopoulou et al., 2023], and EMMA [Yang et al., 2024], as well as advanced VLMs, GPT4O [Hurst et al., 2024a] and LLAVA-13B [Liu et al., 2023b], in the zero-shot setting. Note that EMMA and REFLEXION require successive attempts on the same task. Thus, we report their performance for both 3 and 10 trials.

# 4 Results and Discussion

Table 1 presents ACTIVEVOO's performance against 6 ALFWorld tasks, compares it with 13 baselines representing vision-only, language-only, and vision-language models. ACTIVEVOO achieves an overall success rate of 0.86, the highest among all agents with only visual observations. Our approach significantly outperforms SOTA vision-language models such as GPT-4O and LLAVA-13B in zero-shot settings, and also surpasses models that require extensive task-specific fine-tuning and data collection, including BUTLER, LLAMA-ADAPTER, INSTRUCTBLIP, and EMMA. While EMMA's performance is comparable, it achieves this level after 10 successive trials in the same environment, leveraging cross-trial information transfer. In contrast, ACTIVEVOO is strictly evaluated in a zero-shot setting, based on a single trial, without relying on any prior experience or memory from earlier episodes.

## 4.1 RQ 1: Overall Performance Comparison

Even when compared to language-based methods, which operate on structured textual inputs devoid of visual noise, ACTIVEVOO outperforms all existing LLM-based planners, except for REFLEXION-10, which relies on 10 trials that reflect on past failures in the same environment, while ACTIVEVOO significantly outperforms REFLEXION-3 with 3 trials. Additionally, most existing LLM-based planners require few-shot examples of the same task, which may not be available when the task is new. More impressively, ACTIVEVOO outperforms all approaches that rely on large-scale supervised training or environment-specific fine-tuning, which are often costly and impractical.

Table 1: Success rates on the ALFWorld tasks using template task instructions. "Env." indicates whether the agent uses a visual or textual environment. "Multi-Trail" indicates whether the agent requires multiple trials on the same environment. "Fine-Tuning" states whether the model is fine-tuned with data collected from ALFWorld. "*" indicates that reported results are used, and "–" indicates results are not available.

| Agent | Env. | Multi-Trial | Fine-Tuning | Avg. | Pick | Clean | Heat | Cool | Look | Pick2 |
|---|---|---|---|---|---|---|---|---|---|---|
| Human Performance* [Shridhar et al., 2020] | Visual | ✗ | ✗ | 0.91 | – | – | – | – | – | – |
| **Vision Only Models** | | | | | | | | | | |
| MCNN-FPN* [Shridhar et al., 2020] | Visual | ✗ | ✓ | 0.05 | – | – | – | – | – | – |
| RESNET-18* [Shridhar et al., 2020] | Visual | ✗ | ✓ | 0.06 | – | – | – | – | – | – |
| **Language Only Models** | | | | | | | | | | |
| GPT-4O (Text) [Hurst et al., 2024b] | Textual | ✗ | ✗ | 0.21 | 0.29 | 0.17 | 0.21 | 0.23 | 0.27 | 0.13 |
| REACT [Yao et al., 2022] | Textual | ✗ | ✗ | 0.69 | 0.78 | 0.75 | 0.72 | 0.58 | 0.77 | 0.56 |
| REFLEXION-3 [Shinn et al., 2024] | Textual | ✓(3) | ✗ | 0.83 | 0.89 | 0.80 | 0.78 | 0.85 | 0.86 | 0.81 |
| REFLEXION-10 [Shinn et al., 2024] | Textual | ✓(10) | ✗ | **0.91** | **0.96** | **1.00** | **0.81** | 0.83 | 0.94 | 0.88 |
| DEPS* [Wang et al., 2023] | Textual | ✗ | ✗ | 0.76 | 0.93 | 0.50 | 0.80 | **1.00** | **1.00** | 0.00 |
| AUTOGEN* [Wu et al., 2023] | Textual | ✗ | ✗ | 0.77 | 0.92 | 0.74 | 0.78 | 0.86 | 0.83 | 0.41 |
| **Vision Language Models** | | | | | | | | | | |
| BUTLER* [Shridhar et al., 2020] | Visual | ✗ | ✓ | 0.26 | 0.31 | 0.41 | 0.60 | 0.27 | 0.12 | 0.29 |
| GPT-4O (Vision) [Hurst et al., 2024b] | Visual | ✗ | ✗ | 0.08 | 0.16 | 0.06 | 0.00 | 0.04 | 0.10 | 0.12 |
| LLAVA-13B [Liu et al., 2023b] | Visual | ✗ | ✗ | 0.11 | 0.13 | 0.15 | 0.12 | 0.07 | 0.11 | 0.09 |
| LLAMA-ADAPTER* [Yang et al., 2024] | Visual | ✗ | ✓ | 0.13 | 0.17 | 0.10 | 0.27 | 0.22 | 0.00 | 0.00 |
| INSTRUCTBLIP* [Yang et al., 2024] | Visual | ✗ | ✓ | 0.22 | 0.50 | 0.26 | 0.23 | 0.06 | 0.17 | 0.00 |
| EMMA-3* [Yang et al., 2024] | Visual | ✓(3) | ✓ | 0.37 | 0.55 | 0.41 | 0.45 | 0.13 | 0.65 | 0.4 |
| EMMA-10* [Yang et al., 2024] | Visual | ✓(10) | ✓ | 0.82 | 0.71 | **0.94** | **0.85** | **0.83** | 0.88 | 0.67 |
| (Ours) | Visual | ✗ | ✗ | **0.86** | **0.93** | 0.87 | 0.83 | 0.80 | **0.89** | **0.86** |

For example, BUTLER and EMMA require 52K and 15K expert demonstrations, respectively. In contrast, ACTIVEVOO only requires a high-level symbolic action model typically available for agents with predefined skill sets (see Appendix A in the supplementary material). Given the diversity and variability of open-world task instances, models trained on fixed task distributions often struggle to generalize. In contrast, ACTIVEVOO can reason over any task that can be expressed in PDDL [Haslum et al., 2019] or similar symbolic forms. ACTIVEVOO's strong performance underscores the value of combining structured symbolic reasoning with LLM-inferred commonsense probability estimates, pointing to a promising direction for hybrid planning in embodied AI systems.

## 4.2 RQ 2: Impact of Active Knowledge Acquisition

One key research question we investigate is how much our agent's performance can be credited to active knowledge acquisition (which leverages VOO-inferred objects from visual observations) in comparison to more passive strategies. Under the *exhaustive object acquisition* setup, the agent passively extracts information from each scene by prompting the VLM to extract all possible objects and relations from visual input, without contextual filtering. This method achieves a success rate of only 0.22 as shown in Table 2, likely due to the

Table 2: Impacts of Active Knowledge Acquisition and ActiveVOO Components (RQ2 and RQ3 Results)

| Settings | Success Rate | Episode Length |
|---|---|---|
| ActiveVOO | 0.86 | 15.3 |
| w/ Exhaustive Obj. Acq. | 0.22 | 25.4 |
| w/ Goal-Directed Obj. Acq. | 0.48 | 23.0 |
| w/ LLM Subgoals Obj. Acq. | 0.47 | 22.5 |
| w/o VOO Calculation | 0.75 | 19.2 |
| w/o VOO and Object Partition | 0.68 | 21.3 |
| w/ LLaVA-13B as VLM | 0.82 | 16.1 |
| w/ GPT-3.5 as LLM | 0.77 | 19.5 |

overwhelming number of potential objects and relations present in each image. In the *Goal-Directed Object Acquisition setting*, we instead query the VLM using only the high-level task goal. This leads to an improvement over the fully passive baseline. We also evaluate a strategy where the VLM is queried using a plan generated by an LLM. Both of these weakly guided methods achieve similar success rates (0.47 and 0.48), suggesting that LLM-generated plans alone are insufficient to reliably identify task-relevant objects. Together, these results empirically demonstrate the value of our principled VOO-based approach, which significantly outperforms passive or weakly guided knowledge acquisition strategies. The results are shown in Table 2, and the experimental setup details for these ablations can be found in Appendix G in the supplementary material.

#### 4.2.1 RQ 3: Ablation of ActiveVOO Components

We conduct a detailed ablation study to examine the contributions of various components in the ACTIVEVOO framework as shown in Table 2. We assess the effect of VOO by replacing VOO-selected objects with those chosen solely by highest belief probability. In this setting, if the agent's initial belief is improperly calibrated, it then lacks a mechanism to effectively explore alternatives. This results in a performance drop from $0.86$ to $0.75$ and an increase in the average episode length by $3.9$ steps. We further analyze the performance by removing not only the VOO module but also the mechanism to extract relevant object descriptions. In this case, the agent directly queries the LLM to produce belief probabilities of an entire subgoal. Under this setup, the success rate declines further to $0.68$. We observe that when a subgoal becomes sufficiently complex, i.e., containing many predicates, the LLM's logits become a worse approximation of the agent's actual belief. Lastly, we investigate the effect of using different VLM and LLM models. Replacing the GPT-4o with LLaVA-13B for the VLM component slightly reduces performance to $0.82$, and substituting GPT-4o with GPT-3.5 for the LLM component results in a further drop to $0.77$, indicating that stronger LLM/VLM reasoning capabilities directly enhance ACTIVEVOO's performance.

## 5 Related Work

**Language and Vision Models for Embodied Planning.** Recent work on embodied planning with LLMs focuses on predicting the next action [Ahn et al., 2022, Valmeekam et al., 2023, Hazra et al., 2024] or generating step-by-step plans through prompting [Yao et al., 2022, Shinn et al., 2024, Huang et al., 2022]. Hybrid approaches that integrate classical planners with LLM reasoning have also been proposed [Arora and Kambhampati, 2023, Guan et al., 2023, Hazra et al., 2024], though they typically operate under closed-world assumptions or rely on iterative re-planning. VLM-based methods extend embodied reasoning to visual observations, either via vision-to-language translation [Gao et al., 2024, Huang et al., 2023] or direct visual grounding [Driess et al., 2023, Hurst et al., 2024b]. VLAs based approaches produce low-level action sequences [Kim et al., 2024, Brohan et al., 2022], but generally assume static, fully observable environments. While some open-world VLMs [Yang et al., 2024, Liu et al., 2023b] have been proposed, they rely on costly task-specific fine-tuning and extensive data collection. In contrast, ACTIVEVOO generates lifted plans without assuming full observability or complete knowledge, enabling robust reasoning in open-world settings.

**Active Knowledge Acquisition For Embodied Agents.** Existing works tackle active knowledge acquisition from different perspectives. A notable area of research is embodied navigation, where the agent is tasked with autonomously discovering an object of a certain type [Yang et al., 2018], navigating to a specified image [Zhu et al., 2017], or following step-by-step instructions [Anderson et al., 2018, Ku et al., 2020]. However, these approaches typically assume that the goal or target object is predefined. In contrast, our work focuses on determining object relevance to the overall goal. Other lines of work explore active object recognition [Bohg et al., 2017, Caselles-Dupré et al., 2021] or general exploration of the environment [Zhu et al., 2023b]. While these methods emphasize interaction and perception, they are not task-driven and do not require assessing the relevance of objects, an essential ability for open-world agents. ActiveVOO explicitly addresses this gap by integrating goal-conditioned object relevance estimation with a principled knowledge acquisition framework base on VOO for open-world embodied planning.

## 6 Conclusion

We introduce ACTIVEVOO, a novel framework that enables active knowledge acquisition for open-world embodied agents based on the Value of Observation (VOO). Our method integrates lifted regression for plan subgoal generation with commonsense reasoning and probabilistic inference via LLMs, allowing for structured inference without requiring a complete domain specification. We demonstrate that ACTIVEVOO significantly outperforms state-of-the-art LLM and VLM methods without the need for additional data collection, domain-specific fine-tuning, or few-shot examples.

**Limitation**   This work is evaluated on the ALFWorld simulator, which requires further fine-tuning to transfer effectively to real-world settings involving physical robots. Additionally, it does not address user or contextual preferences in goal specification, which could influence the agent's underlying utility function.

## Acknowledgments

This work was supported by LG Electronics, Toronto AI Lab Grant Ref No. 2024-0565.

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

# A  Action Model

Below are the high-level skills/actions defined in PDDL syntax. Each action $A \in \mathcal{A}$ is represented as

$$A = \{\text{params}(A), \text{pre}(A), \text{add}(A), \text{del}(A)\},$$

corresponding to the action's *parameters*, *preconditions*, and *effects*. The sets $\text{add}(A)$ and $\text{del}(A)$ indicate which predicates are made true or false. Our regression approach requires no negated preconditions or goals, no conditional effects, and no quantifiers in preconditions or effects.

```
(:action PickupObject
  :parameters (?o - obj)
  :precondition (and
    (handEmpty)
    (isObject ?o))
  :effect (and
    (holds ?o)
    (not (handEmpty))))

(:action PutObjectInReceptacle
  :parameters (?o - obj ?r - obj)
  :precondition (and
    (canContain ?r ?o)
    (isObject ?r)
    (holds ?o))
  :effect (and
    (inReceptacle ?o ?r)
    (handEmpty)
    (not (holds ?o)))))

(:action HeatObject
  :parameters (?r - obj ?o - obj)
  :precondition (and
    (isHeatingObject ?r)
    (canHeat ?r ?o)
    (holds ?o))
  :effect (and
    (isHot ?o)))

(:action CleanObject
  :parameters (?r - obj ?o - obj)
  :precondition (and
    (isCleaningObject ?r)
    (canClean ?r ?o)
    (holds ?o))
  :effect (and
    (isClean ?o)))

(:action CoolObject
  :parameters (?r - obj ?o - obj)
  :precondition (and
    (isCoolingObject ?r)
    (canCool ?r ?o)
    (holds ?o))
  :effect (and
    (isCool ?o)))

(:action ToggleObject
  :parameters (?r - obj)
  :precondition (and
    (isObject ?r))
  :effect (and
    (isOn ?r)))
```

# B Open-World Lifted Regression

## B.1 Open-World Planning Formalism

We formalize the open-world planning problem as the tuple $\Pi = \langle \mathcal{P}, \mathcal{A}, O, G, I \rangle$, where $\mathcal{P}$ denotes the set of predicate symbols, $\mathcal{A}$ the set of action schemas, $O$ the (incomplete) set of known objects, $G$ the goal formula, and $I$ the agent's initial knowledge. Unlike closed-world planning, which presumes complete knowledge of all objects and the initial state, open-world agents must accommodate missing object instances as well as uninstantiated variables or predicates. We represent an atom $p(T)$ by pairing a predicate $p \in \mathcal{P}$ with an $n$-tuple of terms $T = \langle t_1, \ldots, t_n \rangle$, where $\mathrm{vars}(T)$ indicates the variables occurring in $T$. An atom is lifted if $\mathrm{vars}(T) \neq \emptyset$. Each action schema $A \in \mathcal{A}$, $A = \{\mathrm{params}(A), \mathrm{pre}(A), \mathrm{add}(A), \mathrm{del}(A)\}$, in which $\mathrm{params}(A)$ are the action's parameters, $\mathrm{pre}(A)$ its preconditions, $\mathrm{add}(A)$ the predicates added to the resulting subgoal, and $\mathrm{del}(A)$ the predicates removed. The objective is to generate a finite set $S = \{(g_i, \pi_i)\}_{i=1}^n$, where each pair $(g_i, \pi_i)$ consists of a feasible lifted *subgoal* $g_i$ and an associated *action sequence* $\pi_i$ whose length does not exceed a user-defined horizon $\tau$.

## B.2 Open-World Lifted Regression

Once the planning problem is formalized, we leverage *Lifted Backward Search* [Ghallab et al., 2004], adapted to open-world settings, to produce a set $S = \{(g_1, \pi_1), \ldots, (g_n, \pi_n)\}$ of all feasible lifted subgoals $g_i$ and their corresponding plans $\pi_i$. Starting from the overall goal $G$, the algorithm iteratively applies REGRESS to each RELEVANT action until no further regression is possible or a predefined length $\tau$ is reached. We say that an action schema $a \in \mathcal{A}$ is RELEVANT to a subgoal $s$ if $s \cap (\mathrm{add}(a) \cup \mathrm{del}(a)) \neq \emptyset$, $s \cap \mathrm{del}(a) = \emptyset$, and $s \cap \mathrm{add}(a) = \emptyset$. Whenever $a$ is relevant, the regressed subgoal is computed as $\mathrm{REGRESS}(s, a) = (s \setminus \mathrm{add}(a)) \cup \mathrm{pre}(a)$. To avoid variable clashes and ensure correctness, each regression step applies the standard logical operations STANDARDIZE, UNIFY, and SUBSTITUTE (see Appendix C). The algorithm maintains a frontier of subgoal-plan pairs $(g, \pi)$, initialized with $(G, \langle \rangle)$. At each iteration, if $|\pi| = \tau$ the current pair is added to $S$ as a terminal subgoal-plan pair; otherwise, for each action $a \in \mathcal{A}$, we STANDARDIZE and UNIFY $a$ with $g$, check RELEVANT$(a, g)$, and enqueue the new pair $\big(\mathrm{REGRESS}(g, a),\ a :: \pi\big)$ if $g$ has not been visited. If no regressible action exists, the pair is deemed terminal and also added to $S$. The process repeats until the frontier is empty, at which point the algorithm returns all lifted plans and subgoals in $S$ of length at most $\tau$. A complete pseudocode listing appears in Appendix C in the supplementary material.

## B.3 Open-World Lifted Regression Algorithm

The Open World Lifted Regression algorithm is initialized with an empty set $S$, which the algorithm will return, and a frontier that contains the overall goal $G$ paired with an empty plan. The algorithm performs a backward search by repeatedly REGRESS through a subgoal and trace pair $(g, \pi)$ from the frontier. If the trace length $|\pi|$ equals the maximum depth $\tau$, it adds $(g, \pi)$ to $S$. For each action schema $A \in \mathcal{A}$, the algorithm applies STANDARDIZE to generate fresh variables to obtain $A'$ and uses UNIFY to match $A'$ effect with the current subgoal $g$. If $g$ is determined to be RELEVANT, the algorithm regresses $g$ through $A'$ to produce a new subgoal $g'$ and an extended trace $\pi.append(A')$. The new subgoal $g'$ is added to the frontier for further regression. If no actions apply, the algorithm treats $(g, \pi)$ as a leaf and adds it to $S$. Once the frontier is exhausted, $S$ contains all lifted subgoals and their corresponding regression paths up to depth $\tau$.

# C Logical Operations

Here is a set of standard logical operations used in this work for Lifted Regression:

- **Substitution**: SUBSTITUTE is an operator that replaces terms in one set of logical statements with terms from another set. We use $\theta$ to represent a mapping between variables/ terms in the form of a dictionary:

$$\theta = \{v_1/t_1, v_2/t_2, v_3/t_3, \ldots, v_n/t_n\}.$$

- **Standardization**: The STANDARDIZE operator, written as STANDARDIZE$(p)$, replaces all variables in a predicate $p$ with new variables $v'$ such that $v' \notin \mathcal{V}$. This ensures that variables in one logical

**Algorithm 1** Open-World Lifted-Regression

```
1: Π = ⟨𝒫, 𝒜, O, G, I⟩ , τ
2: S ← {}
3: Frontier ← {(G, π = [])}
4: Regressed ← {G}
5: while Frontier ≠ ∅ do
6:     g, π ← POP(Frontier)
7:     if len(π) = τ  then
8:         S.add((g, π))
9:     else
10:        RegressActions = {}
11:        for each A in 𝒜 do
12:            A′ ← STANDARDIZE(A)
13:            θ ← UNIFY(A′, g)
14:            if RELEVANT(θ(A′), θ(g)) then
15:                RegressActions.add(A′)
16:                π.append(A′)
17:                g′ ← REGRESS(θ(g), θ(A′))
18:                if g′ not in Visited then
19:                    Regressed.add(g′)
20:                    Frontier.add((g′, π))
21:                end if
22:            end if
23:        end for
24:        if RegressActions = ∅ and g′ ∉ Visited then
25:            S.add((g, π))
26:        end if
27:    end if
28: end while
29: return  S
```

statement do not conflict with those in the target statement it tries to unify. Standardization is carried out by applying substitution $\theta(p)$, which replaces each variable as follows:

$$\forall v \in \text{vars}(p), \quad v \notin \text{vars}(\text{STANDARDIZE}(p)).$$

- **Unification**: $\text{UNIFY}(p, q)$ is an operator that checks whether two logical statements can be made equivalent by applying substitution and standardization to their variables. In Lifted Regression, this is used to match an action with a subgoal. Given two logical statements $p$ and $q$, it returns a substitution $\theta$, which is the Most General Unifier (MGU), such that:

$$\text{UNIFY}(p, q) = \theta \quad \text{where} \quad \theta(p) = \theta(q).$$

## D   Belief MDP Formalism

We cast object-centric active knowledge acquisition as a sequential decision problem: the agent selects which candidate objects to sense to maximize progress toward the goal. We model this as an MDP over the extracted object descriptions $\mathcal{O}$, relational descriptions $\mathcal{R}$, and lifted subgoals $\mathcal{G}$. Formally, we define a Markov Decision Process as a tuple $\mathcal{M} = \langle S, A, T, R \rangle$.

**State Space.**   The state captures verified (true/false) knowledge about objects and relations:

$$S = \big\{ s = (O^+, O^-, R^+, R^-) \mid O^\pm \subseteq \mathcal{O}, \; R^\pm \subseteq \mathcal{R}, \; O^+ \cap O^- = \varnothing, \; R^+ \cap R^- = \varnothing \big\}.$$

Here $O^+$ and $O^-$ are, respectively, the sets of object descriptions verified true or false; $R^+$ and $R^-$ are the analogous sets for relational descriptions. Any description not in $(O^+ \cup O^-) \cup (R^+ \cup R^-)$ is unobserved.

**Action Space.**   At each state, the agent chooses one unobserved description (object or relation) to sense:

$$A(s) = \big\{ \text{SenseObj}(o) \mid o \in \mathcal{O} \setminus (O^+ \cup O^-) \big\} \cup \big\{ \text{SenseRel}(\rho) \mid \rho \in \mathcal{R} \setminus (R^+ \cup R^-) \big\}.$$

Each action corresponds to an information-gathering operation that attempts to confirm whether the chosen description holds in the current environment.

**Transition Model.** The agent may sense an object description $o \in \mathcal{O}$ or a relational description $\rho \in \mathcal{R}$ only if it is currently unobserved. Moreover, a relational sensing action is enabled only when *all* argument-dependent object descriptions of $\rho$ have already been sensed. We assume that the outcome of each sensing action depends only on the corresponding object or relational descriptions rather than the state $s$ and $s'$ in which the sensing occurs.

The transition model is therefore

$$T(s, \texttt{SenseObj}(o), s') = \begin{cases} p_o & s' = (O^+ \cup \{o\}, O^-, R^+, R^-) \\[2mm] 1 - p_o & s' = (O^+, O^- \cup \{o\}, R^+, R^-) \end{cases}$$

$$T(s, \texttt{SenseRel}(\rho), s') = \begin{cases} p_\rho & s' = (O^+, O^-, R^+ \cup \{\rho\}, R^-) \\[2mm] 1 - p_\rho & s' = (O^+, O^-, R^+, R^- \cup \{\rho\}) \end{cases}$$

where $s = (O^+, O^-, R^+, R^-)$, $p_o \in [0, 1]$ is the agent's belief that object $o$ can be sensed, and $p_\rho \in [0, 1]$ is the belief that $\rho$ holds true.

**Reward Function.** The agent receives a terminal reward when its verified knowledge suffices to realize some lifted subgoal. Let $\mathrm{obj}(g) \subseteq \mathcal{O}$ and $\mathrm{rel}(g) \subseteq \mathcal{R}$ denote, respectively, the object and relational descriptions required by $g \in \mathcal{G}$. We define

$$R(s) = \begin{cases} 1, & \exists g \in \mathcal{G} \text{ such that g is satisfiable} \\ 0, & \text{otherwise,} \end{cases} \tag{12}$$

Having defined the MDP, we now describe how the transition probabilities $p_o$ and $p_\rho$ are estimated using a large language model (LLM) to provide commonsense priors over object existence.

# E   Extracting Belief Probability via LLMs

For belief probability extraction from LLM logits, we use OpenAI's API (`GPT-4o`) with a simple `yes`/`no` classification interface. The estimator takes as input an object description in natural language (e.g., "There exists a cooked piece of meat on a plate") and returns a scalar belief probability that reflects the model's confidence in the truth of the statement. To compute this, we prompt the LLM with an existential question and analyze its token-level response. We set the temperature to zero and enable logit tracking.

## E.1   Prompt Construction

We first construct a natural language description of the target object based on its descriptive predicates. Each lifted subgoal generated by regression is represented as a conjunction of unary and binary predicates. After applying object-centric partitioning to extract individual objects from the subgoals, we obtain a set of unary predicates for each object. These predicates are then translated into a coherent natural language phrase that captures the object's type and its functional relationships. The resulting description is embedded into a fixed prompt template that assigns the language model the role of a home assistant robot operating in a household environment.

Below is an example prompt for object description query:

```
You are a home assistant robot operating in a common household environment.
Is it true that you can find an object such that this object is a plate and it is
    clean?

Please answer the question using only ['Yes', 'No']
```

Below is an example prompt for relational description query:

```
You are a home assistant robot operating in a common household environment.

Is it true that object X canclean object Y, given that object X is a cleaning
    object and object Y is a plate?

Please answer the question using only ['Yes', 'No']
```

Each query is repeated 5 times.

# F  Agent Overview

---

**Algorithm 2** ACTIVEVOO Agent

---

**Require:** Instruction $\mathcal{G}_{nl}$, action model $\mathcal{A}$, interaction budget $B$
1: $G \leftarrow$ GOALPARSER$(\mathcal{G}_{nl})$
2: $S \leftarrow$ LIFTEDREGRESSION$(G, \mathcal{A}, \tau)$
3: $\Pi \leftarrow \{\pi \mid \langle g, \pi \rangle \in S\}$
4: $(\mathcal{U}, \mathcal{R}) \leftarrow$ PARTITIONSUBGOALS$(S)$
5: KB $\leftarrow I$; budget $\leftarrow B$
6: **for all** $\langle g, \pi \rangle \in S$ **do**
7: $\quad$ $\mathcal{U}_g \leftarrow$ EXTRACTOBJECTS$(g)$
8: **end for**
9: **while not** SATISFIED$(G, \text{KB})$ **and** budget $> 0$ **do**
10: $\quad$ **p** $\leftarrow \{P(\varphi \mid \pi, \text{KB})\}_{\pi \in \Pi}$
11: $\quad$ **for all** $o \in \mathcal{U}_g$ **do**
12: $\quad\quad$ VOO$(o) \leftarrow$ COMPUTEVOO$(o, \textbf{p})$
13: $\quad$ **end for**
14: $\quad$ $\mathcal{O}_+ \leftarrow \{o \mid \text{VOO}(o) > 0\}$
15: $\quad$ $o^\star \leftarrow \arg\max_{o \in \mathcal{O}_+} \text{VOO}(o)$
16: $\quad$ EXPLORE$(o^\star)$
17: $\quad$ **for all** $o \in \mathcal{O}_+$ **do**
18: $\quad\quad$ OBSERVE$(o)$; UPDATEKB$(o, \text{KB})$
19: $\quad\quad$ budget $\leftarrow$ budget $-1$
20: $\quad\quad$ **if** budget $= 0$ **then**
21: $\quad\quad\quad$ **break**
22: $\quad\quad$ **end if**
23: $\quad$ **end for**
24: $\quad$ **for all** $\langle g, \pi \rangle \in S$ **do**
25: $\quad\quad$ **if** SATISFIED$(g, \text{KB})$ **then**
26: $\quad\quad\quad$ **for all** $a \in \pi$ **do**
27: $\quad\quad\quad\quad$ **if** ACT$(a)$ **succeeds then**
28: $\quad\quad\quad\quad\quad$ PROGRESS$(\text{KB}, a)$
29: $\quad\quad\quad\quad$ **end if**
30: $\quad\quad\quad\quad$ budget $\leftarrow$ budget $-1$
31: $\quad\quad\quad\quad$ **if** budget $= 0$ **then**
32: $\quad\quad\quad\quad\quad$ **break**
33: $\quad\quad\quad\quad$ **end if**
34: $\quad\quad\quad$ **end for**
35: $\quad\quad$ **end if**
36: $\quad$ **end for**
37: **end while**
38: **return** KB

---

For every candidate plan $\pi \in \Pi$, the agent maintains a belief score $P(\varphi \mid \pi, \text{KB})$ that reflects the likelihood that the plan can currently succeed. Using these scores, the agent computes the Value of Observation (VOO) for each object description $o$ extracted from every lifted subgoal $g$. All objects with strictly positive VOO are collected into the set $\mathcal{O}_+$. This positive VOO set represents all observable objects that can potentially improve or alter the agent's decision. The agent then queries the vision–language model for *all* objects in $\mathcal{O}_+$. Each query updates the knowledge base (KB) with new unary and relational facts. After the KB is updated, the agent scans the set of lifted subgoals.

Whenever the predicates of a subgoal $g$ are satisfied, the associated plan $\pi$ becomes executable. The agent immediately executes each action $a \in \pi$, updating the KB after each successful step. The loop terminates when the overall goal $G$ is achieved or the interaction budget is exhausted.

# G Ablations

**Passive and Weakly Guided Baselines:** We evaluate three alternative methods for extracting object information from the environment. *Exhaustive Object Acquisition* is a fully passive approach in which the vision–language model (VLM) is prompted to list all observable objects. *Goal-Directed Object Acquisition* is a weakly guided approach that prompts the VLM using only the overall task goal. *Plan-Directed Object Acquisition* first prompts the VLM to generate a plan and then queries for all objects mentioned in that plan. The exact prompts used in our experiments are provided below.

- **Exhaustive Object Acquisition:**

```
You are a home assistant robot operating in a common household environment
Give me all objects you can observe from the scene.

Please answer in this format: ["obj_1", "obj_2", ..., "obj_n"]
```

- **Goal-Directed Object Acquisition:**

```
You are a home assistant robot operating in a common household environment
Give me all objects you can observe from the scene that are related to the goal
    {GOAL}.

Please answer in this format: ["obj_1", "obj_2", ..., "obj_n"]
```

- **LLM-Subgoal Object Acquisition:**
  - *Extract high-level plan*

```
You are a home assistant robot operating in a common household environment
Your goal is to {GOAL}.

Give me a plan that can achieve the goal.
```

  - *Extract objects from the plan:*

```
You are a home assistant robot operating in a common household environment
Your plan is {PLAN}.

Give me all objects you can observe from the scene that are related to your
    plan.

Please answer in this format: ["obj_1", "obj_2", ..., "obj_n"]
```

All extracted object names are post-processed to align with the vocabulary used in ALFWorld; however, the list of ALFWorld object names is never exposed during extraction. We employ `GPT-4o` as the VLM in all of these experiments.

**Component Ablations within ACTIVEVOO:** We next disable or swap individual components of ACTIVEVOO:

- **No VOO:** Without VOO, the agent extracts objects only from the plan it believes is most likely to succeed, that is, $p = \arg\max_{p'} P(p')$, where the belief probabilities are obtained from the LLM. We retain the same object-partition method but limit extraction to the chosen plan $p$.
- **No VOO + No Partition:** VOO calculation requires partitioning objects by their first-order descriptions. When partitioning is disabled, we cannot compute per-object plan probabilities, so we ask the LLM for the feasibility of the entire plan. The prompt is:

```
You are a home assistant robot operating in a common household environment
Is the plan {PLAN} feasible to execute in this environment?

Please answer using only ['Yes', 'No'].
```

- **VLM Swap:** We replace GPT-4o with LLaVA-13B for vision–language object extraction while still using GPT-4o to compute belief probabilities.
- **LLM Swap:** We substitute GPT-4o with GPT-3.5 Turbo for belief-probability estimation, keeping GPT-4o as the VLM for visual information extraction.

