# OpenReview forum: "ActiveVOO: Value of Observation Guided Active Knowledge Acquisition for Open-World Embodied Lifted Regression Planning"
_NeurIPS.cc/2025/Conference — NeurIPS 2025 poster_

### Official Review · Reviewer_CBjT · 2025-06-13

**Clarity:** 4
**Significance:** 3
**Originality:** 3
**Rating:** 5
**Confidence:** 4

**Summary:**

This paper proposes a novel method named ActiveVOI for embodied task planning with open-world and zero-shot capabilities. ActiveVOI uses Lifted Regression to generate subgoals which then help identify task-relevant objects. Then a quantification of object utility named Value of Information (VOI) is introduced, which is generated by LLMs and VLMs. Experiments on this framework are carried out on ALFWorld benchmark, showing its outperformance over state-of-the-art LLM/VLM based planners.

**Questions:**

1. Will there be experiments with other embodied environments like Habitat and iGibson?

2. Will there be experiments under real-world settings?

3. Is there any measure taken to overcome LLM’s problems of inconsistency and prompt-dependency, and/or help correct possible bias?

**Ethical Concerns:**

["NO or VERY MINOR ethics concerns only"]

**Final Justification:**

As my concerns are well addressed, I am increasing my score

**Limitations:**

Yes.

**Paper Formatting Concerns:**

None.

**Quality:**

3

**Strengths And Weaknesses:**

Strengths：

1. This paper presents its novel method with detailed explanation and illustration, along with clear and rigorous mathematical descriptions.

2. This paper presents an organic combination of Lift Regression and VOI theory for open-world active knowledge acquisition, and cleverly makes good use of commonsense and reasoning abilities of LLMs/VLMs.

3. Experiments prove this framework’s SOTA performance under ALFWorld environment, and multiple ablation studies clearly present the contribution of each component including VOI calculation and subgoal object acquisition.

Weaknesses：

1.  The VOI calculation highly relies on LLM and VLMs, which we know face the problems of inconsistency and prompt-dependency, especially when the scene is complicated, or the instruction is unclear.

2. The experiment is conducted on ALFWorld exclusively, which lacks validation on generalizability.

3. Some might have concerns about novelty, as notions of active knowledge acquisition and VOI are already established, and this paper just makes combination in a new context.

---

> ### Author Rebuttal · Authors · 2025-07-30
>
> ## Rebuttal
>
> ### \[Weakness 1, Question 3]: LLM/VLM Consistency and Sensitivity
>
> **Reviewer comment:**
> >“The VOI calculation highly relies on LLM and VLMs, which we know face the problems of inconsistency and prompt-dependency, especially when the scene is complicated, or the instruction is unclear.”
>
> >“Is there any measure taken to overcome LLM’s problems of inconsistency and prompt-dependency, and/or help correct possible bias?”
>
> **Response:**
>
> We appreciate the reviewer’s concern about the reliability of LLM and VLM outputs in complex scenes and under prompt variability. Below, we describe the specific design choices we implemented to mitigate inconsistency, reduce prompt sensitivity, and ensure robustness in ActiveVOI’s commonsense reasoning pipeline.
>
> **Regarding LLM/VLM Prompt Sensitivity and Inconsistency:** To mitigate prompt sensitivity and reduce reliance on domain-specific prompt engineering, we adopt three key design choices.
>
> * We kept the prompts minimal and generic without additional task-specific prompt tuning or templates (shown in Appendix D.1).
> * We leverage test-time scaling techniques of self-consistency to improve reliability of probability estimates by aggregating LLM outputs 10 times, resulting in a more stable estimation of commonsense priors.
> * ActiveVOI operates entirely in a zero-shot setting, avoiding prompt-induced bias and ensuring better generalization across unseen tasks.
>
> **On Handling Complicated Scene:** We handle complex scenes by reducing the object search space through compact lifted object descriptions and by removing duplication of semantically equivalent objects. These strategies can reduce LLM/VLM inconsistencies by limiting queries to a small set of goal-relevant objects, rather than reasoning over all observed entities.
>
> **On Handling Unclear Instructions or Goals:** Unclear instructions or ill-defined goals pose fundamentally different challenges in open-world planning that require separate methodologies and benchmarks. Prior work frames these problems through contingent planning \[1] or preference elicitation \[2]. While important, these issues are orthogonal to our focus on object-centric knowledge acquisition in open-world planning.
>
> ---
>
> ### \[Weakness 2, Questions 1, 2]: Motivation for ALFWorld Compared to Other Benchmarks
>
> **Reviewer comment:**
> >“The experiment is conducted on ALFWorld exclusively, which lacks validation on generalizability.”
>
> **Reviewer questions:**
> >“Will there be experiments with other embodied environments like Habitat and iGibson?”
>
> >“Will there be experiments under real-world settings?”
>
> **Response:**
>
> We thank the reviewer for raising the important question of generalizability. Below, we clarify our motivation for choosing ALFWorld over other embodied benchmarks and explain the current limitations of evaluating in real-world settings.
>
> **Habitat, iGibson Feasibility:** While Habitat and iGibson are popular embodied AI benchmarks, they are primarily designed for object navigation and rearrangement tasks. In such settings, object relevance is trivial, as all necessary object knowledge is directly provided. As a result, these benchmarks are not suitable for evaluating open-world planning and active sensing methods, where agents must infer which objects are relevant under partial knowledge and semantic uncertainty.
>
> **Motivation for Using ALFWorld:** We selected ALFWorld tasks because they have several key characteristics essential for our open-world active sensing evaluation requirements:
> (1) task-relevant objects are not directly given and require semantic inference;
> (2) the environment features high diversity in objects, scenes, and tasks, along with rich visual observations;
> (3) it is partially observable and can be configured without access to ground-truth object and relational information; and
> (4) the benchmark poses significant challenges for SOTA models (VLMs), even with extensive domain-specific fine-tuning (shown in Table 1).
>
> **On Real-World Evaluation:** To the best of our knowledge, there is currently no real-world setup that can support the level of object diversity, action complexity, with object state changes required to test open-world planning and active knowledge acquisition. Building such a testbed would require significant engineering effort, which we view as outside the core contributions of our work which is our ActiveVOI framework based on formal decision-theoretic reasoning.
>
> ---
>
> ### \[Weakness 3]: Regarding Novelty of ActiveVOI
>
> **Reviewer comment:**
> >“Some might have concerns about novelty, as notions of active knowledge acquisition and VOI are already established, and this paper just makes a combination in a new context.”
>
> **Response:**
>
> We would like to clarify that ActiveVOI is not merely a combination of existing approaches, but addresses an overlooked and fundamental challenge in open-world planning: (1) determining which objects are relevant for sensing, and (2) estimating the expected utility of sensing those objects. This problem is particularly difficult in open-world settings, where the agent operates under partial observability and lacks prior knowledge about object types, configurations, and affordances, while the set of potentially observable objects is unbounded. In contrast, most existing work on active perception assumes that the set of relevant objects is given. Thus, ActiveVOI fills this important but often overlooked gap in existing active sensing approaches \[3].
>
> ---
>
> ### References
>
> \[1] Dearden, Richard, et al. "Incremental contingency planning." 13th International Conference on Automated Planning and Scheduling. 2003.
>
> \[2] Ren, Allen Z., et al. "Robots That Ask For Help: Uncertainty Alignment for Large Language Model Planners." Conference on Robot Learning. PMLR, 2023.
>
> \[3] Crespo, Jonathan, et al. "Semantic information for robot navigation: A survey." Applied Sciences 10.2 (2020): 497

---

> ### Author Response · Authors · 2025-08-06
>
> Dear reviewer:
>
> We sincerely appreciate your insightful review. With the reviewer-author discussion period ending soon, please let us know if we can further clarify our responses to any of your concerns.
>
> best,
>
> The Authors

---

> > ### Comment · Reviewer_CBjT · 2025-08-09
> > **Thank you for response**
> >
> > Thank you for detailed reply. The concerns regarding LLM/VLM and benchmark selection are well clarified. As for the novelty, other reviewers also raised the concern regarding other task planning methods. The open-vocabulary setting is also explored in the LLM-based task planning works which leverages the open-voc object detection model for recognition. The novelty should be enhanced and further explained.
> >
> > Meanwhile, the real-world robot experiment is also desired according to other reviewers, as there are many mobile manipulation works conducting real-robot experiments.

---

> > > ### Author Response · Authors · 2025-08-09
> > >
> > > We sincerely thank the reviewer for acknowledging our clarifications on the LLM/VLM components and benchmark selection. Below, we want to address the comments on novelty and real-world experiments.
> > >
> > > ### Novelty in active sensing
> > > Most prior active sensing works target *object* or *semantic navigation*, where the objective is (1) to locate a pre-specified object and (2) to maximize *information gain*.
> > > In contrast, **ActiveVOI** addresses two underexplored problems:
> > > 1. **What to sense** – determining which type of object should be sensed in the first place.
> > > 2. **Utility-aware sensing** – evaluating how much *utility* (not just information) is gained from sensing a specific object in relation to the current plan.
> > >
> > > These two aspects constitute novel contributions beyond the conventional active sensing paradigm.
> > >
> > > ### Novelty in task planning
> > > We are, to the best of our knowledge, the first to combine **LLM commonsense reasoning** with **lifted regression planning** for open-world *active exploration*.
> > > This combination enables:
> > > 1. Avoiding the combinatorial explosion of object–relation properties in open-world settings.
> > > 2. Generating sound and verifiable plans without requiring a complete initial knowledge state.
> > > 3. Greatly improving reliability by avoiding reliance on LLMs to generate large domain descriptions or to reason over long horizons with many action constraints.
> > >
> > > ### Difference from open-vocabulary object detection
> > > We see **open-vocabulary** object detection** as *parallel* to our work, rather than solving the same problem.
> > > OV detection answers questions like *“find an object of type: drink”*, whereas our framework asks *“how relevant is object X (e.g., microwave, a sink) to my goal of putting a hot egg on a plate?”*.
> > > We address this relevance estimation through a formal lifted regression framework coupled with LLM commonsense and VOI-based reasoning.
> > >
> > > ### On real-world robotics and manipulation
> > > To the best of our knowledge, there is currently no real-world setup supporting the diversity of objects, action complexity, and object state changes required to test open-world planning with active knowledge acquisition.
> > > Building such a testbed would require significant engineering effort, which is outside the scope of this work.
> > > Our contribution is the **ActiveVOI** framework itself, grounded in formal decision-theoretic reasoning.
> > >
> > > We hope these clarifications address your concerns and we remain grateful for your constructive feedback.

---

### Official Review · Reviewer_H9Rd · 2025-06-24

**Clarity:** 2
**Significance:** 3
**Originality:** 3
**Rating:** 3
**Confidence:** 3

**Summary:**

This paper introduces ActiveVOI, a framework for open-world embodied planning under partial observability and incomplete knowledge. The key innovation is an active, object-centric approach to knowledge acquisition: instead of passively gathering all available information, the agent uses Lifted Regression to generate compact, relational subgoal descriptions and identifies relevant objects via the theory of Value of Information (VOI). Commonsense priors from LLMs/VLMs help estimate object utility. Experimental validation is performed on the visual ALFWorld benchmark, showing substantial improvements over previous (even fine-tuned) LLM/VLM-based baselines.

**Questions:**

- Are there concrete differences between using LLMs for task-relevant object prediction/information gain and previous LLM- or information-theoretic approaches to active perception?
- Do the authors anticipate that this approach would transfer smoothly from simulated benchmarks (ALFWorld) to real-world robot settings? If not, why they choosing the ALFWorld simulator rather than other Open-World Embodied tasks?
- One of the claims of the framework is its incorporation of LLMs/VLMs in quantifying the utility of knowledge acquisition targets—i.e., using LLM-based commonsense for Value of Information (VOI) calculations and prioritization. If a traditional symbolic/planning agent were given the same—fully specified and correct—knowledge base (i.e., a list of all objects, their types, and relations), could it not also generate an optimal plan using standard planning techniques (such as PDDL/STRIPS)? In other words, is the reliance on LLM commonsense truly necessary for high performance, or is it mainly compensating for incomplete or noisy object knowledge?

**Ethical Concerns:**

["NO or VERY MINOR ethics concerns only"]

**Final Justification:**

While the paper demonstrates significant strengths, there remain a few points that I am not fully convinced by like the limited evaluating pipeline (only tested on a single benchmark modified by the authors) and the "open-world" setting argued by the author (which I think is "partial observability" but not really "open-world").

**Limitations:**

See ABOVE.

**Quality:**

3

**Strengths And Weaknesses:**

Strengths:
- The paper tackles a crucial challenge in embodied AI: practical and scalable planning in open worlds, where passive information gathering is infeasible due to combinatorial complexity, although it's not a new concept.
- The proposed framework combines Lifted Regression for global, object-centric reasoning with a decision-theoretic (VOI) framework for prioritizing which objects to sense.
- The formalization of open-world planning with lifted predicates, explicit attention to handling incomplete knowledge, and proof-based (sound and complete) guarantees for the regression process. VOI application to active perception/planning is theoretically justified and well-motivated.

Weaknesses:
- The writing and the figures of the manuscript requires further improvement. For example, fig.1 shows complex pipelines, while the differences of the two pipelines only mainly lines in the Active Knowledge Acquisition step.
- ALFWorld is not an Open-World Embodied setting, at least from my point of view. There are some similar works tackling the active reasoning problems in environments like Minecraft or Crafter, which I think is more suitable to call as open-world. The author should consider testing their algrithims in more scenarios.
-  Active perception and information-driven exploration are long-standing topics in AI and robotics, especially using VOI. Recent works have begun to use LLMs (and even VLMs) for goal decomposition, subgoal extraction, and commonsense reasoning for embodied agents. I'm not sure whether the integration of VOI with LLMs for active object-centric search significantly novel, or just an incremental composition of existing tools.

---

> ### Author Rebuttal · Authors · 2025-07-30
>
> We sincerely thank the reviewer for the detailed and thoughtful feedback. Below, we address the concerns and questions raised.
>
> ## \[Weakness 1] Regarding Presentation and Clarity
>
> **Reviewer comment:**
>
> > "The writing and the figures of the manuscript requires further improvement. For example, Fig. 1 shows complex pipelines, while the differences of the two pipelines only mainly lines in the Active Knowledge Acquisition step."
>
> **Response:**
> We thank the reviewer for their feedback on Figure 1. Our intention was to illustrate the difference between 1) passive and active knowledge acquisition and 2) grounded progression and lifted regression. We can certainly revise the figure to improve clarity; to help us improve it, please let us know which of the following distinctions depicted in the figure seems unclear:
>
> * **Difference between planning strategy (Steps 4,6 Left, Steps 8,9 Right):** Grounded progression replans if actions fail or when knowledge is insufficient for plan generation. In contrast, lifted regression generates a compact set of feasible lifted plans up front.
> * **Difference in Knowledge Base Size (KB blocks):** Passive sensing requires the agent to store all object and relational information, resulting in a much larger KB than active sensing, which stores only task-relevant information.
> * **Active Perception Pipeline (Step 5 Left):** The active method uses lifted subgoals and the agent’s knowledge base to selectively acquire task-relevant information from observations.
>
> ## \[Weakness 2] "Open-World" Clarification
>
> **Reviewer comment:**
>
> > "ALFWorld is not an Open-World Embodied setting ... similar works tackling the active reasoning problems in environments like Minecraft or Crafter ... The author should consider testing their algrithims in more scenarios."
>
> **Response:**
>
> **Open-World Clarification:** Our use of the term open-world follows the Open World definition from a model-theoretic perspective, where the agent operates with “incomplete knowledge of the model” \[1], i.e., incomplete information about object types, configurations, and relational properties.
>
> **Minecraft-based benchmark:** Most Minecraft-based benchmarks are considered open-world due to their expansive environments and open-ended task structures. However, they are highly domain-specific and offer limited generalizability, as they focus on high-level actions like “mining” and “crafting” over a fixed set of predefined objects and materials unique to the Minecraft domain \[2]. Minecraft recipes are also widely available in LLM training data, removing the need to infer new object types, relations, or affordances, which are core challenges in open-world planning and active sensing.
>
> **ALFWorld benchmark:** We configure ALFWorld to align with our definition of open-world settings, where agents operate under partial observability using only egocentric RGB input, without access to ground-truth object types, attributes, or relational information. This makes it well-suited for evaluating open-world active sensing. A comparison with other embodied AI domains is provided in the next section (see \[Weakness 3, Question 1]).
>
> ## \[Weakness 3, Question 1] Novelty with Comparison of Existing Approaches
>
> **Reviewer comment:**
>
> > "Active perception and information-driven exploration are long-standing topics in AI and robotics ... LLMs (and even VLMs) for goal decomposition, subgoal extraction ... not sure whether the integration of VOI with LLMs for active object-centric search significantly novel..."
> >
> > "Are there concrete differences between using LLMs for task-relevant object prediction/information gain and previous LLM- or information-theoretic approaches to active perception?"
>
> **Response:**
>
> ActiveVOI introduces a novel formulation of active sensing by explicitly reasoning over goal-driven object relevance and utility in the open-world setting. Here, we highlight the novelty of ActiveVOI with a comparison with existing active sensing work.
>
> **Novelty of ActiveVOI:** Most existing work on active sensing focuses on object or semantic navigation, where the goal is to (1) locate a known object, and (2) to maximize information gain \[6]. In contrast, ActiveVOI tackles important but underexplored problems in active sensing: (1) determining what type of object the agent should sense, and (2) assessing how much utility (not just information) is gained from sensing an object. These two components constitute novel contributions to the field.
>
> **Comparison with LLM/VLM subgoal decomposition:** LLM/VLM subgoal decomposition suffers from several known issues: (1) lacking any correctness guarantee of generated plans, (2) requiring few short examples, (3) poor alignment with the environment and agent’s actions, (4) performance degradation over long-horizon. In contrast, ActiveVOI is based on a formal decision-theoretic framework of lifted regression that: (1) guarantees correctness of the plan, (2) can plan in a zero-shot setting, (3) is based on the agent’s action model, and (4) is capable of tracking long-horizon reasoning.
>
> **Comparison with Information-theoretical approaches:** Traditional information-theoretical approaches often define utility as generic information gain (such as entropy reduction) without directly connecting this gain to the agent’s goal \[7]. In contrast, ActiveVOI is task-centric, where the utility is defined with respect to goal achievement. This distinction is crucial in goal-oriented tasks, where not all information is equally valuable, and task relevance must guide sensing.
>
> ## \[Question 2] Real-World Feasibility and ALFWorld Choice
>
> **Reviewer comment:**
>
> > "Do the authors anticipate that this approach would transfer smoothly from simulated benchmarks (ALFWorld) to real-world robot settings? If not, why they choosing the ALFWorld simulator rather than other Open-World Embodied tasks?"
>
> **Response:**
>
> We want to clarify how the method can support real-world deployment as a high-level reasoning module, and justify our choice of ALFWorld over other embodied AI benchmarks in the context of open-world active sensing.
>
> **Real-world transfer:** The primary obstacles for the transfer of ActiveVOI lie in integrating low-level control and hardware-specific tuning. ActiveVOI, as a reasoning and planning module, should be readily adaptable to real-world applications due to (1) being zero-shot without the need for fine-tuning, (2) avoiding the combinatorial explosion of objects by using a compact lifted-representation of relevant objects, which increase scalability.
>
> **Suitable Embodied AI benchmarks:** We considered several other benchmarks based on Habitat, iGibson, and VirtualHome \[3, 4, 5], but found them insufficient for evaluating open-world planning and active sensing. Tasks designed on these benchmarks are focused on 1) object navigation, 2) object rearrangement, 3) instruction following where all relevant objects are assumed known (given in the instructions), making semantic reasoning and object-centric active sensing irrelevant.
>
> **ALFWorld Rationale:** We selected ALFWorld tasks because they have several key characteristics essential for our open-world active sensing evaluation requirements: (1) task-relevant objects are not directly given and require semantic inference; (2) the environment features high diversity in objects, scenes, and tasks, along with rich visual observations; (3) it is partially observable and can be configured without access to ground-truth object and relational information; and (4) the benchmark poses significant challenges for SOTA models (VLMs), even with extensive domain-specific fine-tuning (show in Table 1).
>
> ## \[Question 3] Could a Classical Planner Succeed with Full KB? Is LLM Commonsense Really Needed?
>
> **Reviewer comment:**
>
> > "One of the claims  ... If a traditional symbolic/planning agent ... could it not also generate an optimal plan ... is the reliance on LLM commonsense truly necessary ... or is it ... for incomplete or noisy object knowledge?"
>
> **Response:**
>
> LLMs and VLMs are used not only for knowledge completion of the agent’s KB, but also to estimate the non-deterministic outcomes of sensing actions. As a result, a plan generated by a standard PDDL model, even with a fully specified and correct KB, may not be optimal, as it typically assumes deterministic outcomes. ActiveVOI uses LLM-derived commonsense priors to approximate the transition probabilities of sensing actions (the likelihood of encountering a relevant object (e.g., a heat source) in a given environment (e.g., a kitchen). This allows the agent to compute the expected utility of sensing different objects under our Value of Information framework. In summary, LLMs and VLMs are essential in our framework, both for completing missing knowledge and for modelling the non-determinism of sensing actions.
>
> ## References
>
> \[1] Finzi, Alberto, Fiora Pirri, and Raymond Reiter. "Open world planning in the situation calculus." AAAI/IAAI. 2000.
>
> \[2] Wang, Zihao, et al. "Describe, explain, plan and select: interactive planning with llms enables open-world multi-task agents." Advances in Neural Information Processing Systems (2023)
>
> \[3] Puig, Xavier, et al. "Virtualhome: Simulating household activities via programs." Proceedings of the IEEE conference on computer vision and pattern recognition. 2018.
>
> \[4] Savva, Manolis, et al. "Habitat: A platform for embodied ai research." Proceedings of the IEEE/CVF international conference on computer vision. 2019.
>
> \[5] Shen, Bokui, et al. "igibson 1.0: A simulation environment for interactive tasks in large realistic scenes." International Conference on Intelligent Robots and Systems (IROS) 2021.
>
> \[6] Crespo, Jonathan, et al. "Semantic information for robot navigation: A survey." Applied Sciences 2020
>
> \[7] Stachniss, Cyrill, Giorgio Grisetti, and Wolfram Burgard. "Information gain-based exploration using rao-blackwellized particle filters." Robotics: Science and systems  2005.

---

> > ### Comment · Reviewer_H9Rd · 2025-08-06
> >
> > Thank you very much for the authors’ response. I greatly appreciate the detailed rebuttal, which has addressed some of my concerns. I have carefully reconsidered my scores.
> >
> > While the paper demonstrates significant strengths, there remain a few points that I am not fully convinced by, even after considering the authors’ rebuttal. Specifically, I still believe that the proposed method, as tested on the ALFWorld environment, may not truly constitute an "open-world" setting—even with the modifications mentioned by the authors. The agent operates under partial observability, yet the underlying set of objects, possible relations, and the task space, remains closed and pre-specified. The original ALFWorld benchmark also doesn't argue it as an open-world setting. The experimental results also seems insufficiently with the proposed algorithm only tested in a single scenario. As reviewer z6oJ said, testing in different environments would help to confirm whether the improvement are consistent across environments and tasks.

---

> > > ### Author Response · Authors · 2025-08-06
> > > **Open World Clarification**
> > >
> > > We thank the reviewer for the thoughtful feedback and genuinely appreciate the points raised. However, we believe there are important misunderstandings regarding the term **“open world”** and the rationale behind using the **ALFWorld** benchmark that we would like to clarify further.
> > >
> > > ---
> > >
> > > ### **Regarding Open World**
> > > We follow the **decision‑theoretic definition** of “open world” [1]: *an agent must make decisions with incomplete knowledge of the model*. This notion is well‑established in planning [1,2,3] and robotics [4,5,6] and aligns with the **Open‑World Assumption (OWA)** in logic theory.
> > >
> > > While some work conflates **“open world”** with **“open‑world games,”** the two are fundamentally different. For example, in the prominent Minecraft‑based work DEPS [7], the authors clarify in a footnote on the first page:
> > > > *“We borrow the term ‘open world’ from the game community. It highlights that the agent can navigate inside a diverse environment and accomplish open‑ended tasks freely.”*
> > >
> > > We therefore believe it would be unfair to judge our work based on the **“open‑world games”** definition rather than the long‑standing definition in planning and robotics.
> > >
> > > ---
> > >
> > > ### **Regarding ALFWorld and Other Baselines**
> > > We considered benchmarks such as VirtualHome [8] and Habitat [9], but found them insufficient due to their emphasis on instruction following without the need for state tracking:
> > >
> > > For example, a typical VirtualHome task are given instruction like:
> > > > *“Grab a potato, heat it with the microwave, and put the potato on the plate.”*
> > >
> > > This explicitly lists all relevant objects (“potato,” “microwave,” “plate”), leaving little need for object‑centric knowledge acquisition.
> > >
> > > In contrast, an ALFWorld task such as:
> > > > *“Put a hot potato on the plate”*
> > >
> > > requires the agent to infer that the potato must be **hot** and that it must locate and use a **heating device** to achieve this state.
> > >
> > > We emphasize that our choice of ALFWorld is motivated by its alignment with the **open‑world** decision‑theoretic setting (without any object type, relation and task information), where agents must infer missing knowledge rather than simply execute high-level instructions.
> > >
> > > ---
> > >
> > > ### **References**
> > > [1] Finzi, Alberto, Fiora Pirri, and Raymond Reiter. "Open world planning in the situation calculus." *AAAI/IAAI*. 2000.
> > > [2] Kautz, H., and Selman, B. 1996. Pushing the envelope: Planning, propositional logic and stochastic search. In *Proceedings of the National Conference on Artificial Intelligence (AAAI’96)*.
> > > [3] Smith, D., and Weld, D. 1998. Conformant graphplan. In *Proceedings of the National Conference on Artificial Intelligence (AAAI’98)*, 889–896. AAAI Press/MIT Press.
> > > [4] Jiang, Yuqian, et al. "Open-world reasoning for service robots." *Proceedings of the International Conference on Automated Planning and Scheduling*. Vol. 29. 2019.
> > > [5] Hanheide, Marc, et al. "Robot task planning and explanation in open and uncertain worlds." *Artificial Intelligence* 247 (2017): 119-150.
> > > [6] Talamadupula, Kartik, et al. "Integrating a closed world planner with an open world robot: A case study." *Proceedings of the AAAI Conference on Artificial Intelligence*. Vol. 24. No. 1. 2010.
> > > [7] Wang, Zihao, et al. "Describe, explain, plan and select: interactive planning with large language models enables open-world multi-task agents." *Proceedings of the 37th International Conference on Neural Information Processing Systems*. 2023.
> > > [8] Puig, Xavier, et al. "Virtualhome: Simulating household activities via programs." *Proceedings of the IEEE Conference on Computer Vision and Pattern Recognition*. 2018.
> > > [9] Savva, Manolis, et al. "Habitat: A platform for embodied AI research." *Proceedings of the IEEE/CVF International Conference on Computer Vision*. 2019.

---

> > > > ### Comment · Reviewer_H9Rd · 2025-08-07
> > > >
> > > > Thank you for the detailed clarification regarding the "open world" definition and your rationale for choosing ALFWorld. I greatly appreciate the authors taking the time to address these points with specific references and examples.
> > > >
> > > > I understand that there are multiple valid perspectives on what constitutes an "open world" setting, and I acknowledge that your use of the decision-theoretic definition from planning literature is well-founded. This may only reflect my particular perspective on what constitutes "open world" and sufficient evaluation. For example, regarding your illustration ("Grab a potato, heat it with the microwave, and put the potato on the plate" vs "Put a hot potato on the plate"), I believe this is more fundamentally related to commonsense reasoning rather than open-world characteristics. The challenge here involves understanding implicit action sequences and world knowledge as you mentioned.
> > > >
> > > > Anyway, thank you again for the further clarification.

---

> > > > > ### Author Response · Authors · 2025-08-07
> > > > >
> > > > > We thank the reviewer for acknowledging that our definition of “open world” is well-founded. We sincerely appreciate the reviewer’s insightful feedback and found the discussion intellectually stimulating. We will make sure to incorporate our discussion into the discussion section of the paper.
> > > > >
> > > > > Regarding to
> > > > >
> > > > > > "I believe this is more fundamentally related to commonsense reasoning rather than open-world characteristics. The challenge here involves understanding implicit action sequences and world knowledge as you mentioned."
> > > > >
> > > > > We just like to point out that the ability to reason with **commonsense world knowledge** and to **plan based on implicit action sequences** are arguably what enable us humans (the most intelligent agents we know) to make effective decisions in the real world, which is the ultimate open world. :)

---

### Official Review · Reviewer_z6oJ · 2025-07-02

**Clarity:** 4
**Significance:** 2
**Originality:** 4
**Rating:** 3
**Confidence:** 3

**Summary:**

The paper presents ACTIVEVOI, a novel framework for active knowledge acquisition in open-world embodied planning. It addresses the challenges faced by AI agents in environments with partial observability and incomplete task-relevant knowledge. ACTIVEVOI introduces a two-step approach: (1) determining relevant object descriptions through Lifted Regression, and (2) quantifying and prioritizing knowledge acquisition based on the Value of Information (VOI) theory, using commonsense knowledge from large language and vision-language models (LLMs/VLMs). The framework is evaluated on the ALFWorld benchmark, showing substantial improvements over existing methods, especially in visual tasks. ACTIVEVOI does not require task-specific fine-tuning and demonstrates superior performance in zero-shot settings.

**Questions:**

Could the authors provide further details on how the method might scale in real-world environments? Specifically, how might the approach be adapted to handle environments with significantly more objects or dynamic interactions between objects?

How does the proposed VOI calculation handle dependencies between object attributes and relations in more complex environments, where the assumptions of independence may not hold? Would relaxing these assumptions improve performance?

Can the authors elaborate on potential strategies for reducing the computational cost of VOI calculations, especially when scaling the method to larger environments with many objects?

**Ethical Concerns:**

["NO or VERY MINOR ethics concerns only"]

**Final Justification:**

The applicable scenarios of the method have obvious limitations. Based on the current results, I am very concerned about the practical value of the method.

**Limitations:**

Yes, the authors have addressed the limitations of their work, particularly in terms of the evaluation environment and the theoretical assumptions made. However, there could be more discussion on the real-world applicability of the framework and how it can be adapted to handle more complex, dynamic environments.

**Paper Formatting Concerns:**

The paper format complies with the standards.

**Quality:**

3

**Strengths And Weaknesses:**

Strengths:

Novelty: ACTIVEVOI offers a fresh approach to active knowledge acquisition by combining Lifted Regression with VOI, addressing a crucial gap in open-world embodied planning.

Effectiveness: The framework demonstrates significant improvements over existing models, particularly in zero-shot settings where no prior data is available for training or fine-tuning. This is a noteworthy achievement in open-world environments, which are inherently challenging.

Practical Application: The integration of commonsense knowledge from LLMs/VLMs enables the agent to make intelligent decisions on what information to acquire, potentially leading to more efficient planning in real-world applications.

Weaknesses:

Limited Benchmark Evaluation: The experiments are primarily conducted on the ALFWorld benchmark, which is a well-regarded testing environment, but additional evaluations across other benchmark tasks, such as Alfred or VirtualHome would help to confirm whether the observed performance improvements are consistent across different types of environments and tasks. Expanding the evaluation scope would provide more robust evidence of the framework's general applicability.

Assumptions on Knowledge Independence: The framework makes several assumptions regarding the independence of object attributes and relational properties. In complex real-world scenarios with intricate object interactions, these assumptions may lead to inaccuracies in the VOI calculations, affecting the agent's decision-making process.

Scalability and Efficiency: While the method is effective in small-scale environments, it remains unclear how well the system scales to much larger, more dynamic real-world environments with a high number of objects and complex interrelationships.

---

> ### Author Rebuttal · Authors · 2025-07-30
>
> ## Rebuttal
>
> ### \[Weakness 1] Benchmark Choice
>
> **Reviewer comment:**
>
> > "The experiments are primarily conducted on the ALFWorld benchmark, which is a well-regarded testing environment, but additional evaluations across other benchmark tasks, such as Alfred or VirtualHome would help to confirm whether the observed performance improvements are consistent across different types of environments and tasks. Expanding the evaluation scope would provide more robust evidence of the framework's general applicability."
>
> **Response:**
>
> We thank the reviewer for highlighting the importance of evaluating generalizability across diverse embodied benchmarks. However, we find most benchmarks inadequate for our evaluation purposes. Below, we explain our rationale for focusing on ALFWorld and clarify the limitations of alternative environments for evaluating open-world planning and object-centric knowledge acquisition.
>
> **ALFRED and VirtualHome:**
> We considered both ALFRED and VirtualHome, but found them unsuitable for evaluating semantic open-world planning and object-centric active knowledge acquisition. Tasks in both environments assumes subgoals or step-by-step instructions are available to the agent which already contain all necessary objects required to achieve the goal. This makes object-centric active sensing unnecessary.
>
> **ALFWorld:**
> We selected ALFWorld because it exhibits key characteristics essential for our evaluation:
>
> 1. Task-relevant objects are not directly specified and require semantic inference.
> 2. The environment features high diversity in objects, scenes, and tasks, with rich visual observations.
> 3. It is partially observable and can be configured without access to ground-truth object types or relations.
> 4. The benchmark remains challenging for state-of-the-art models (e.g., VLMs), even with extensive domain-specific fine-tuning (as shown in Table 1).
>
> ---
>
> ### \[Weakness 2, Question 2] Assumptions on Object Knowledge Independence
>
> **Reviewer comment:**
>
> > "The framework makes several assumptions regarding the independence of object attributes and relational properties. In complex real-world scenarios with intricate object interactions, these assumptions may lead to inaccuracies in the VOI calculations, affecting the agent's decision-making process."
> >
> > "How does the proposed VOI calculation handle dependencies between object attributes and relations in more complex environments, where the assumptions of independence may not hold? Would relaxing these assumptions improve performance?"
>
> **Response:**
>
> We agree that assumptions of independence must be handled carefully. While we discussed the motivation in Section 2.1.3, we provide further elaboration below that we will include in the revision.
>
> **1. Justification for the Independence Assumption**
>
> * **Reduce sensing targets:** By assuming independence between object attributes and relations, we enable object-level sensing, drastically reducing the number of sensing candidates. Without this assumption, the agent would have to reason over a joint space of all object-relation configurations, which is computationally intractable at scale.
> * **Enable reliable LLM reasoning:** LLMs are more reliable at generating commonsense priors for concise object descriptions (e.g., "a knife is likely in the kitchen") than for complex relational queries (e.g., "is it likely to find a dirty and metal knife near a sponge with a dirty plate?"). As shown in the "w/o VOI and Object Partition" row in Table 2, reasoning over complex subgoals leads to a significant performance drop (−18%), emphasizing the value of object-level prioritization.
>
> **2. Robustness to Biased VOI Calculation**
> While initial VOI estimates may be biased due to the independence assumption and model limitations, the agent's belief state is updated incrementally with new observations. VOI is re-computed over this updated KB, allowing recalibration and alignment with the actual environment over time. This dynamic adjustment mitigates the impact of initial bias.
>
> ---
>
> ### \[Weakness 3, Questions 1 & 3] Scalability and Cost
>
> **Reviewer comment:**
>
> > "Could the authors provide further details on how the method might scale in real-world environments? Specifically, how might the approach be adapted to handle environments with significantly more objects or dynamic interactions between objects?"
> >
> > "While the method is effective in small-scale environments, it remains unclear how well the system scales to much larger, more dynamic real-world environments with a high number of objects and complex interrelationships."
> >
> > "Can the authors elaborate on potential strategies for reducing the computational cost of VOI calculations, especially when scaling the method to larger environments with many objects?"
>
> **Response:**
>
> **Handling Environments with Many Objects:**
> We thank the reviewer for raising the important issue of scalability, which is one of the core motivations behind ActiveVOI. Real-world environments may contain thousands of objects, yet only a small fraction is typically relevant for a given goal. ActiveVOI is expressly designed to exploit this observation:
>
> * **Lifted relational abstraction:** Our planner uses lifted regression to perform goal-directed backward search, deriving relevant subgoal descriptions that filter out irrelevant objects.
> * **Object description consolidation:** When multiple subgoals share object descriptions, we consolidate them to eliminate redundant sensing queries, further improving efficiency.
>
> These strategies ensure that the agent’s sensing and reasoning complexity scales with the number of goal-relevant objects, not the total number of visible objects.
>
> **Reducing VOI Computation Cost:**
> The above design choices greatly reduce VOI computational cost by narrowing the search space. Another major contributor to cost is LLM query overhead. ActiveVOI minimizes this cost through:
>
> * **Symbolic planner offloading:** Planning is delegated to a symbolic planner, whose computation is negligible compared to LLMs.
> * **LLM token efficiency:** ActiveVOI uses \~2,000 LLM tokens per task, compared to >50,000 in step-by-step LLM planning methods like ReAct.
>
> **Handling Dynamic Environments:**
> We acknowledge that adapting to dynamic or probabilistic settings is an important future direction. However, doing so would require different methodologies and baselines (e.g., conformant or FOND planning \[2,3]). These settings are orthogonal to our current focus, but we will include discussion of this in the final version.
>
> ---
>
> **References:**
>
> \[1] Valmeekam, Karthik, et al. "On the planning abilities of large language models—a critical investigation." *NeurIPS* (2023).
>
> \[2] Hoffmann, Jörg, and Ronen I. Brafman. "Conformant planning via heuristic forward search: A new approach." *Artificial Intelligence* 170.6-7 (2006): 507-541.
>
> \[3] Muise, Christian, Sheila McIlraith, and Christopher Beck. "Improved non-deterministic planning by exploiting state relevance." *ICAPS* (2012).

---

> ### Author Response · Authors · 2025-08-06
>
> Dear reviewer:
>
> We sincerely appreciate your insightful review. With the reviewer-author discussion period ending soon, please let us know if we can further clarify our responses to any of your concerns.
>
> best,
>
> The Authors

---

> > ### Comment · Reviewer_z6oJ · 2025-08-09
> >
> > Thanks for the detailed reply. However, these weaknesses are related to the quality of the original paper. I'm open to discussing it with fellow reviewers.

---

> > > ### Author Response · Authors · 2025-08-09
> > >
> > > Dear Reviewer,
> > >
> > > Thank you for your reply to our rebuttal. We are doing our best to address all questions and concerns raised in your review. As the review period is coming to a close, please let us know if there are any points that you feel remain insufficiently addressed, and we will make every effort to clarify them promptly.
> > >
> > > Sincerely,
> > > The Authors

---

### Official Review · Reviewer_YV56 · 2025-07-05

**Clarity:** 3
**Significance:** 3
**Originality:** 4
**Rating:** 5
**Confidence:** 3

**Summary:**

This paper introduces ActiveVOI, a zero-shot framework for open-world embodied planning that actively acquires task-relevant information using Lifted Regression and Value of Information (VOI). The agent identifies subgoals via symbolic regression, uses LLM/VLM commonsense to estimate object relevance, and then prioritizes sensing based on VOI. On the ALFWorld benchmark, ActiveVOI achieves strong performance, outperforming most baselines in a single-trial visual setting, without any fine-tuning.

**Questions:**

- **LLM/VLM failure handling**: In cases where the LLM or VLM provides incorrect or misleading priors (e.g., low belief for a relevant object), does ActiveVOI have any fallback or exploration mechanism? Could belief calibration, ensembling, or conservative VOI strategies improve robustness to noisy commonsense reasoning?

- **Knowledge persistence and reuse**: Have you considered augmenting ActiveVOI with a memory system for storing verified object properties or relations across episodes? This could support lifelong learning and reduce redundant sensing in related future tasks. Would VOI still be usable in such a setting?

**Ethical Concerns:**

["NO or VERY MINOR ethics concerns only"]

**Final Justification:**

My questions and concerns have been addressed. With the inclusion of an illustrative example showing how the proposed approach, ActiveVOI, mitigates initial commonsense errors, as well as a discussion on scalability and long-term knowledge persistence as future work, I believe this work merits acceptance.

**Limitations:**

yes.

**Paper Formatting Concerns:**

no formatting issues.

**Quality:**

3

**Strengths And Weaknesses:**

Strengths:
- **Strong empirical performance in single-trial, zero-shot settings**: ActiveVOI achieves top-tier success rates among all visual agents that operate in a single trial without fine-tuning, significantly outperforming state-of-the-art baselines like GPT-4O (vision), LLAVA-13B, and BUTLER. This highlights the practicality and robustness of the method in realistic, one-shot planning scenarios.

- **Principled and novel framework**: The combination of Lifted Regression for generating symbolic subgoals and VOI-guided active sensing represents a creative and theoretically well-grounded integration of symbolic AI and modern pretrained models. The decomposition into object-centric descriptions and relational constraints allows for efficient and targeted exploration.

- **Clear methodology with strong theoretical foundation**: The paper includes rigorous formalization of the planning setup, VOI estimation, and regression mechanics. The object-relevance partitioning and the assumptions underlying the belief factorization are clearly articulated.


Weaknesses:
- **Heavy reliance on LLM/VLM accuracy**: The method assumes that GPT-4 (or similar models) provides reliable estimates for object relevance. As shown in ablations, performance drops with weaker LLMs (e.g., GPT-3.5). This dependence may limit robustness, especially when commonsense priors are uncertain or incorrect—an issue not deeply explored in the failure analysis.

- **Scalability and memory handling remain open questions**: The planning horizon is capped, and the system operates episodically without a persistent knowledge base. In lifelong or multi-task settings, the agent would likely re-discover the same information repeatedly, as no explicit mechanism exists for long-term memory, knowledge abstraction, or reuse.

---

> ### Author Rebuttal · Authors · 2025-07-30
>
> We thank the reviewer for the detailed and insightful feedback. Below, we address the reviewer’s concerns and questions regarding LLM/VLM accuracy, failure recovery, and the potential for knowledge persistence and reuse.
>
> ## \[Weakness 1, Question 1] Reliance on LLM/VLM Accuracy and Failure Handling
>
> **Reviewer comments:**
>
> > “The method assumes that GPT-4 (or similar models) provides reliable estimates for object relevance. As shown in ablations, performance drops with weaker LLMs (e.g., GPT-3.5). This dependence may limit robustness, especially when commonsense priors are uncertain or incorrect—an issue not deeply explored in the failure analysis.”
>
> > “LLM/VLM failure handling: In cases where the LLM or VLM provides incorrect or misleading priors (e.g., low belief for a relevant object), does ActiveVOI have any fallback or exploration mechanism? Could belief calibration, ensembling, or conservative VOI strategies improve robustness to noisy commonsense reasoning?”
>
> **Response:**
>
> **Reliance on LLM/VLM accuracy:**
> We appreciate the reviewer’s concern regarding the reliance on LLM/VLM estimates for object relevance. In fact, ActiveVOI is not statically bound to these initial priors. The agent maintains a knowledge base (KB) that is incrementally updated with each new observation, and VOI is re-computed over this updated belief state. As the agent gathers more information, its KB progressively aligns with the actual environment. This incremental recalibration of VOI helps mitigate the impact of initial commonsense errors.
>
> **Fallback mechanisms and potential for improving robustness:**
> While the current version of ActiveVOI does not implement a fallback mechanism, this was a deliberate design choice to isolate and evaluate the contribution of VOI-driven acquisition. That said, we included ablations against three alternative acquisition strategies that may be seen as surrogates for a fallback mechanism: Exhaustive, Goal-directed, and LLM-Subgoal-guided acquisition (Table 2), all of which show over 40% performance degradation. The ablations suggest that ActiveVOI is already comparatively more robust than other strategies. We agree that robustness could be further improved by integrating fallback strategies, especially in safety-critical or real-world applications. One potential direction is to incorporate belief calibration or LLM/VLM uncertainty thresholds to trigger a conservative default strategy when model confidence is low.
>
> We appreciate the reviewer’s comments on these important issues and will highlight this discussion in the revised manuscript.
>
> ---
>
> ## \[Weakness 2, Question 2] Scalability and Memory Persistence
>
> **Reviewer comment:**
>
> > “Scalability and memory handling remain open questions: The planning horizon is capped, and the system operates episodically without a persistent knowledge base. In lifelong or multi-task settings, the agent would likely re-discover the same information repeatedly, as no explicit mechanism exists for long-term memory, knowledge abstraction, or reuse.”
>
> > “Knowledge persistence and reuse: Have you considered augmenting ActiveVOI with a memory system for storing verified object properties or relations across episodes? This could support lifelong learning and reduce redundant sensing in related future tasks. Would VOI still be usable in such a setting?”
>
> **Response:**
>
> We thank the reviewer for raising the important points of planning horizon selection, scalability, and long-term knowledge persistence. Below, we clarify how the current ActiveVOI framework addresses these issues and highlight future extensions.
>
> **Planning horizon:**
> The lifted-regression planner used in ActiveVOI requires a finite planning horizon $t$ to enumerate all lifted plans up to depth $t$. In practice, $t$ is chosen based on the upper limit of the planning horizon reflecting time and cost constraints. For all benchmarks in our paper, we set $t = 15$, which corresponds to the upper bound of the expected episode length.
>
> **Per-episode KB and VOI caching:**
> ActiveVOI maintains a structured KB during each episode. The KB compactly represents all verified object properties and relations as grounded predicates and is continuously updated through the agent’s observations and interactions. Additionally, we cache computed VOI scores to prevent redundant reasoning, which improves both cost and efficiency.
>
> **Persistent memory and VOI for lifelong/multitask settings:**
> Supporting persistent memory across episodes could be an interesting future direction. Unlike LLM-based approaches that store full interaction trajectories, our framework enables structured transfer via compact symbolic KBs (and optionally their VOI caches). These representations are lightweight, abstract away irrelevant task-specific details, and can be easily reused across tasks in a lifelong learning setting.
>
> The primary adaptation required to support VOI in lifelong settings is to provide a mechanism for Bayesian updating (recalibration) of object sensing probabilities in light of previous sensing experience. Existing comparison benchmarks do not evaluate this setting, but we will discuss this important future work direction in the revision.

---

> > ### Comment · Reviewer_YV56 · 2025-08-07
> >
> > Thank you for the detailed response, which has addressed most of my questions and concerns. It would be helpful to include an example to support the claim that “this incremental recalibration of VOI helps mitigate the impact of initial commonsense errors.” If such an example is already included in the paper, I would appreciate it if you could kindly direct me to it. Based on the above considerations, I will maintain my current rating.

---

> > > ### Author Response · Authors · 2025-08-08
> > >
> > > We thank the reviewer for their thoughtful feedback and suggestions. The paper currently do not include a standalone example specifically illustrating VOI recalibration in response to an initial false belief from the LLM. This is because we did not consider this one of the core contributions of this work. That said, Figure 2 (page 4) does show how VOI is dynamically recalculated (see the blue VOI calculation block) based on the agent’s current knowledge base, observations, and LLM-inferred beliefs (Steps 5 and 6). We agree with the reviewer that including a concrete example of this behavior would help clarify the mechanism, and we will add such an illustration in the final version of the paper.

---

> ### Author Response · Authors · 2025-08-06
>
> Dear reviewer:
>
> We sincerely appreciate your insightful review. With the reviewer-author discussion period ending soon, please let us know if we can further clarify our responses to any of your concerns.
>
> best,
>
> The Authors

---

### Note · Authors · 2025-08-12

**Dear AC,**

First, we thank all reviewers for their thoughtful feedback. We believe the weaknesses and questions from the initial reviews were adequately addressed in our rebuttal. Below, we provide a summary to **additional questions** raised during the discussion.

---

### **Definition of “Open World” (Reviewer YV56)**

In our response to Reviewer YV56, we clarified that our work focuses on “open world” in the **decision-theoretic** sense, where agents:
- *make decision with incomplete knowledge of the model.*

This contrasts with the notion of “open world” in **open-world games** (e.g., Minecraft), where agents:
- *conduct open-ended tasks in diverse environments.*

Our "open world" definition aligns with established works in planning and robotics. We thank Reviewer YV56 for acknowledging that our definition is well-founded, and we are commit clarify the distinction in the final draft.

---

### **Explanation of Novelty (Reviewer CBjT)**

Our work is novel in its contributions to both **active sensing** and **task planning**.

### **Novelty in Active Sensing**
Most prior work focuses on:
1. Locate a pre-specified object type, or
2. Maximize information gain.

In contrast, **ActiveVOI** addresses:
- **What to sense** – identifying sensing targets from feasible plans.
- **Utility gain** – assessing the utility, not just information, from sensing targets.

### **Novelty in Task Planning**
Our method uniquely combine **LLM commonsense** with **lifted regression planning** for open-world sensing and planning, enabling:
- Object-level knowledge acquisition under decision-theoretic lifted regression
- Avoidance of combinatorial explosion in object–relation properties
- More effect and robust planning than LLM/VLM based methods

Our results show  **significant zero-shot performance gains** over methods fine-tuned on domain-specific data.

---

### **Real-World Robotics Setups (Reviewer CBjT)**

We emphasize that our contribution is the **ActiveVOI** framework, which focuses on **task-level** open-world planning and active knowledge acquisition, rather than low-level manipulation or navigation.

Our evaluation spans diverse tasks in visually complex environments and demands long-horizon reasoning over object state changes without predefined object types or action affordances. As evidenced by our baseline results, this setting remains extremely challenging for both **state-of-the-art symbolic** and **LLM/VLM-based** open-world planners.

---

### Decision · Program_Chairs · 2025-09-17

**Decision:**

Accept (poster)

**Comment:**

Initial reviews for this work were generally positive-mixed (1 accept, 2 borderline accept, 1 borderline reject). Key concerns were: (i) unclear technical contributions given reliance on established ideas like VOI and lifted regression with LLM priors, (ii) evaluation limited to ALFWorld with no test of generalization to other embodied environments, (iii) lack of prompt-sensitivity and failure mode analysis for LLM outputs, (iv) limited discussion of scalability and memory handling for long-term knowledge persistence in complex environments, and (v) unclear positioning relative to prior work, making it hard to attribute gains to the core contribution vs. stronger LLM/VLM priors.

The authors' response was thorough and provided experiments and clarifications to address many of these concerns. However, the reviewers remained split, with 2 "Accepts" and 2 "Borderline Rejects." The remaining concerns were more about the limited evaluation (Reviewers H9Rd and z6oJ) and terminology around "open world" (Reviewers H9Rd). After reviewing the paper, reviews, and the authors' response, the AC feels that the paper makes a good contribution whose strengths outweigh the weaknesses and recommends acceptance. The authors are strongly recommended to incorporate the discussion from the rebuttal into the final version for completeness.